# YtfK activates the stringent response by triggering the alarmone synthetase SpoT in *Escherichia coli*

Elsa Germain [1]*, Paul Guiraud[1], Deborah Byrne[2], Badreddine Douzi[1,3], Meriem Djendli[1] & Etienne Maisonneuve[1]*

The stringent response is a general bacterial stress response that allows bacteria to adapt and survive adverse conditions. This reprogramming of cell physiology is caused by the accumulation of the alarmone (p)ppGpp which, in *Escherichia coli*, depends on the (p)ppGpp synthetase RelA and the bifunctional (p)ppGpp synthetase/hydrolase SpoT. Although conditions that control SpoT-dependent (p)ppGpp accumulation have been described, the molecular mechanisms regulating the switching from (p)ppGpp degradation to synthesis remain poorly understood. Here, we show that the protein YtfK promotes SpoT-dependent accumulation of (p)ppGpp in *E. coli* and is required for activation of the stringent response during phosphate and fatty acid starvation. Our results indicate that YtfK can interact with SpoT. We propose that YtfK activates the stringent response by tilting the catalytic balance of SpoT toward (p)ppGpp synthesis.

[1] Laboratoire de Chimie Bactérienne, Institut de Microbiologie de la Méditerranée, CNRS-Aix Marseille Univ (UMR7283), Marseille, France. [2] Protein Expression Facility, Institut de Microbiologie de la Méditerranée, CNRS-Aix Marseille Univ, Marseille, France. [3] Present address: Université de Lorraine, Inra, DynAMic, F-54000 Nancy, France. *email: egermain@imm.cnrs.fr; emaisonneuve@imm.cnrs.fr

Fast and robust adaptive responses are critical for life in harsh environments. Therefore bacteria, including major pathogens, have evolved molecular mechanisms that allow them to respond and to adapt to their surrounding environment. In particular, the ubiquitous stringent response is a general bacterial stress response induced by diverse nutritional and environmental stresses that switches the metabolism balance from growth and division to stress response and survival. This global resetting of bacterial cell physiology relies on the accumulation of the alarmones ppGpp (guanosine 5′-diphosphate, 3′-diphosphate) and pppGpp (guanosine 5′-triphosphate, 3′-diphosphate) [collectively named (p)ppGpp][1].

In Gram-negative bacteria, (p)ppGpp reprograms transcription via binding to RNA polymerase, thereby altering its promoter selectivity[2–5]. Indeed, (p)ppGpp influences the transcription rates of around 500 genes in *E. coli*[6,7]. In addition to transcriptional control, (p)ppGpp directly inhibits several enzymes, including DNA primase, translation factors, polyphosphate kinase, and lysine decarboxylase[8].

The RelA–SpoT Homologue (RSH) family of bifunctional proteins constitutes the key players in synthesizing and degrading (p)ppGpp[9]. Therefore, the balance between both activities represents a critical point of regulation to control the intracellular level of (p)ppGpp. Long RSH proteins have two functional regions: the enzymatic N-terminal half (NTD), encompassing the synthetase (SYNTH) and the hydrolase (HD) domains, and the C-terminal regulatory half (CTD)[10,11]. In most gamma and betaproteobacteria, including *E. coli*, the stringent response is controlled by two RSHs called RelA and SpoT. SpoT has both (p)ppGpp hydrolytic and synthetic activities similar to the Rel enzymes, the long bifunctional RSH enzyme present in the vast majority of bacteria[12–14]. However, despite the strong homology between RelA and SpoT, RelA maintains a pseudo-hydrolase domain that is structurally conserved but nevertheless enzymatically inactive[15–17]. Thus, RelA is a monofunctional (p)ppGpp synthetase. Amino acid starvation that leads to loading of RelA•tRNA complexes at the ribosomal A-site activates RelA's (p)ppGpp synthetic activity in vivo and in vitro[18,19]. In contrast, SpoT functions as a hub protein that integrates various environmental signals, including fatty acid (FA)[20], carbon[12], iron[21], and phosphate starvation[22]. Importantly, the hydrolysis function of SpoT is crucial for balancing cellular (p)ppGpp concentration in the presence of RelA and disruption of *spoT* in wild-type *E. coli* strain is lethal due to the toxic accumulation of (p)ppGpp[12]. Hence SpoT-dependent regulation of intracellular (p)ppGpp level can reflect changes in either synthetase or hydrolase activity. How the switching between the two enzymatic activities is regulated remains largely unknown despite intensive investigations. However, control of SpoT activities through protein–protein interaction plays an important role. Indeed, the acyl carrier protein (ACP) binds SpoT and triggers (p)ppGpp synthesis in response to FA starvation[23]. More recently, the anti-$\sigma^{70}$ factor Rsd of *E. coli* was shown to interact directly with SpoT to regulate stringent response during carbon source downshift by stimulating (p)ppGpp hydrolysis[24].

Here, we present a genetic assay for the identification of protein candidates that can modulate SpoT-dependent (p)ppGpp synthesis in *E. coli*. Overproduction of one of these proteins, YtfK, promotes accumulation of (p)ppGpp in vivo. YtfK is required to maintain elevated (p)ppGpp levels in response to phosphate and FA starvation, and enhances survival and promotes formation of antibiotic tolerant cells under nutritional stress conditions. YtfK appears to interact with SpoT. We propose that YtfK acts by tilting the catalytic balance of SpoT toward (p)ppGpp synthesis rather than hydrolysis.

## Results

### YtfK promotes SpoT-dependent accumulation of (p)ppGpp.
We designed a simple genetic assay allowing us to identify protein partners in *E. coli* that potentially modulate SpoT-dependent accumulation of (p)ppGpp. We exploited the fact that growth in the presence of 1 mM of serine, methionine, and glycine (SMG medium), which induces isoleucine starvation, requires elevated levels of (p)ppGpp[25]. Indeed, expression of isoleucine biosynthetic operon is stimulated by (p)ppGpp and, as a result, a *ΔrelA* mutant is unable to grow on SMG plates (Fig. 1a)[25]. To identify potential activators of SpoT-dependent (p)ppGpp synthesis, we selected for genes that, in multiple copies, would suppress the growth defect of an MG1655 *E. coli* K-12 *ΔrelA* strain on SMG agar plates. We pooled a collection of plasmids obtained from the ASKA library containing almost all *E. coli* K-12 genes[26], each cloned into the high-copy-number vector pCA24N downstream of the isopropyl β-D-1-thiogalactopyranoside (IPTG)-inducible $P_{T5-lac}$ promoter, and identified several putative candidate genes as described in the Methods section. Among them, *ytfK* suppressed the growth defect of a *ΔrelA* mutant on SMG medium. This result was further confirmed after re-cloning the coding region of *ytfK* (without any tag) in a more suitable physiological plasmid harboring a tightly IPTG-inducible $P_{T5-lac}$ promoter (pEG25) (Fig. 1b; Supplementary Fig. 1a, b). Interestingly, *ytfK* encodes a small protein of 8 kDa and is expressed during phosphate starvation[27], a condition known to trigger SpoT-dependent accumulation of (p)ppGpp[22].

To assess the role of YtfK in the stringent response, we monitored the (p)ppGpp level after ectopic *ytfK* expression in a *ΔrelA* strain. As shown in Fig. 1c, the (p)ppGpp level increased dramatically 15 min after *ytfK* induction. Importantly, overexpression of *ytfK* failed to suppress amino acid auxotrophic phenotype characteristic of ppGpp$^0$ strain (*ΔrelA spoT* mutant) while *relA* expression complements the growth defect, showing that YtfK is not a small alarmone synthetase (Supplementary Fig. 1a). Taken together our results show that YtfK triggers SpoT-dependent accumulation of (p)ppGpp in *E. coli*.

### The SpoT-YtfK ratio controls the switch of SpoT activities.
Interestingly, we did not select the *spoT* allele in the multicopy suppressor assay. To determine whether SpoT level is involved in the regulation of (p)ppGpp synthetase and hydrolase activities, we monitored the growth of a *ΔrelA* mutant carrying plasmid pEG25-*spoT* on SMG plates. As shown in Fig. 1b, induction of *spoT* failed to suppress the growth defect of a *ΔrelA* mutant on SMG medium. This result supports the notion that ectopic production of SpoT is not associated with an increase level of (p)ppGpp. Importantly, we observed that induction of *spoT* above 50 μM of IPTG rendered WT strain unable to grow on SMG plates (Supplementary Fig. 1a). Moreover, strong induction of *spoT* triggered amino acid auxotrophic phenotype in *ΔrelA* mutant (Supplementary Fig. 1a). These results strongly support that high level of SpoT reduces basal level of (p)ppGpp, consistent with previous observations[13]. Finally, we observed that induction of *spoT* complements the amino acid auxotrophic phenotype of the ppGpp$^0$ strain only at concentration below 250 μM IPTG. Taken together, these data suggest that artificial control of SpoT level determines the switching from (p)ppGpp synthesis to degradation.

Our observation that ectopic production of YtfK triggers SpoT-dependent (p)ppGpp accumulation in absence of external stresses (Fig. 1c) raises the possibility that the regulation of SpoT activity by YtfK is likely to be acting by a change in the SpoT/YtfK ratio.

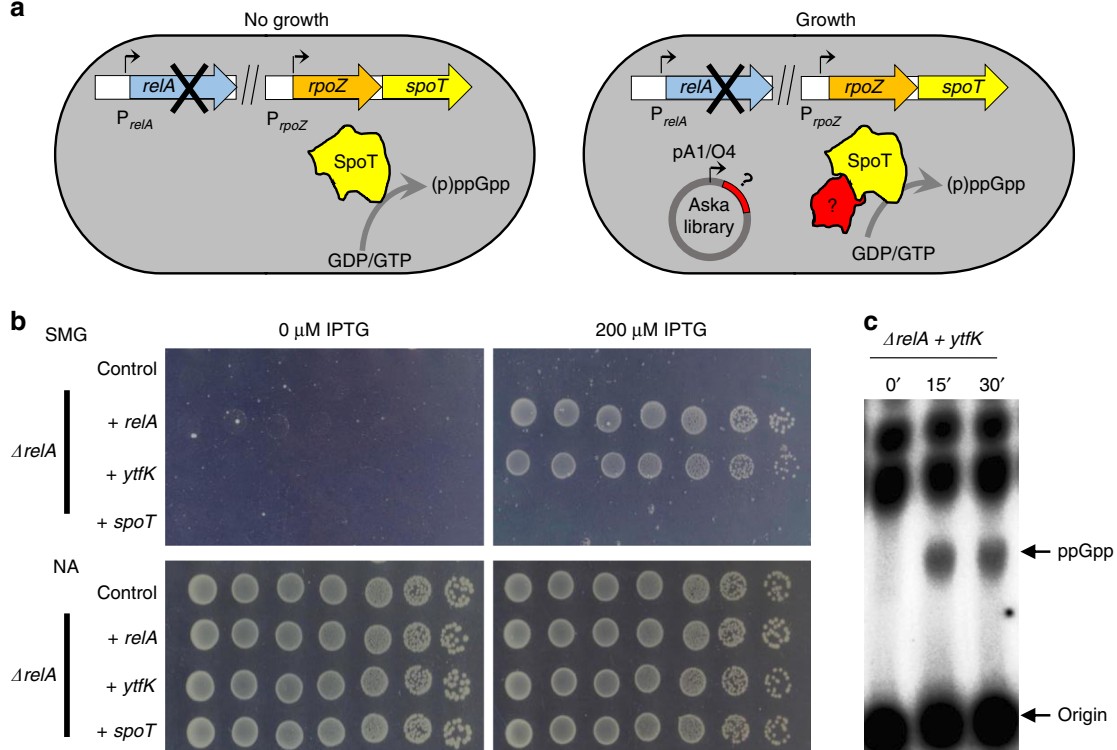

**Fig. 1 YtfK promotes SpoT (p)ppGpp synthesis. a** Genetic setup based on the multicopy suppression (ASKA library) of the nongrowing phenotype of Δ*relA* mutant on SMG plates. SMG plate induces isoleucine starvation and requires high (p)ppGpp level to de-repress isoleucine biosynthesis operon (see the Methods section). Question mark denotes putative candidate that promotes SpoT-dependent (p)ppGpp accumulation. **b** *ytfK* overexpression suppresses the nongrowing phenotype of the Δ*relA* mutant on SMG plates. WT and Δ*relA* mutant were transformed with pEG25 harboring either *relA*, *ytfK*, or *spoT* under an IPTG-inducible promoter. Cells were serial diluted and spotted both on nutrient agar (NA) and SMG plates with or without IPTG. This experiment was repeated five times with identical results. Additional controls and concentration of IPTG are provided in Supplementary Fig. 1. **c** In vivo (p)ppGpp accumulation following ectopic expression of *ytfK*. Δ*relA* mutant carrying *ytfK* on pEG25 was grown exponentially in phosphate MOPS minimal medium (see the Methods section). Samples were collected before and after *ytfK* induction (1 mM IPTG) prior to nucleotide extraction and separation by TLC. Representative autoradiograph of the TLC plates is shown. This experiment was repeated three times with equivalent pattern of (p)ppGpp accumulation. Source data are provided as a Source Data file.

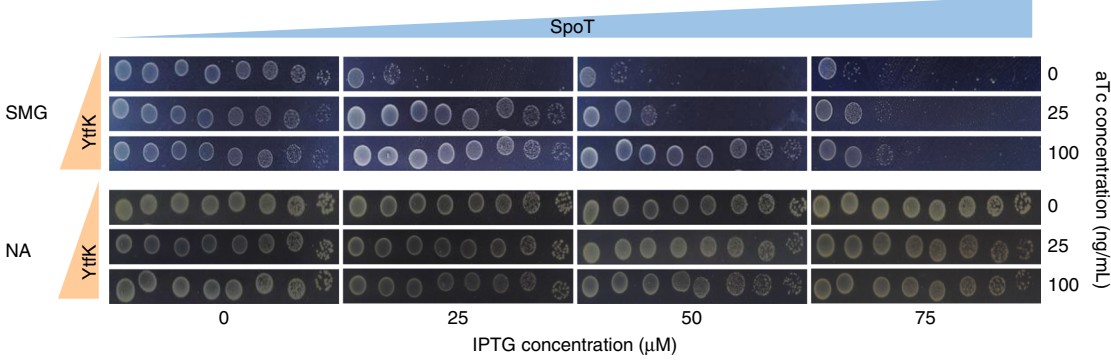

**Fig. 2 The YtfK/SpoT ratio determines growth on SMG plates.** WT strain was co-transformed with pEG25 harboring *spoT* gene under an IPTG-inducible promoter and with pBbS2K harboring *ytfK* gene under an anhydrotetracyclin (aTc) promoter. Cells were serially diluted and spotted both on SMG and NA medium with gradual concentration of IPTG (to induce *spoT*) and aTc (to induce *ytfK*). Additional controls with empty plasmids are provided in Supplementary Fig. 2. Experiments have been repeated four times with similar results.

To address this possibility, we thoroughly monitored the growth phenotype of the WT strain on SMG plates as function of SpoT and YtfK levels. As observed in Fig. 2 (and in Supplementary Fig. 2), a gradual induction of SpoT rendered WT cells unable to grow on SMG plate. Remarkably, this growth defect is shifted toward higher concentration of SpoT when YtfK is increased (Fig. 2; Supplementary Fig. 2). Therefore, the SpoT-YtfK ratio

controls the switching from (p)ppGpp degradation to synthesis. Taken together, our results show that SpoT synthetase activity is subjected to YtfK limitation in vivo.

**YtfK is required in response to phosphate and FA starvation.** Previous work suggested that the transcriptional activator of the

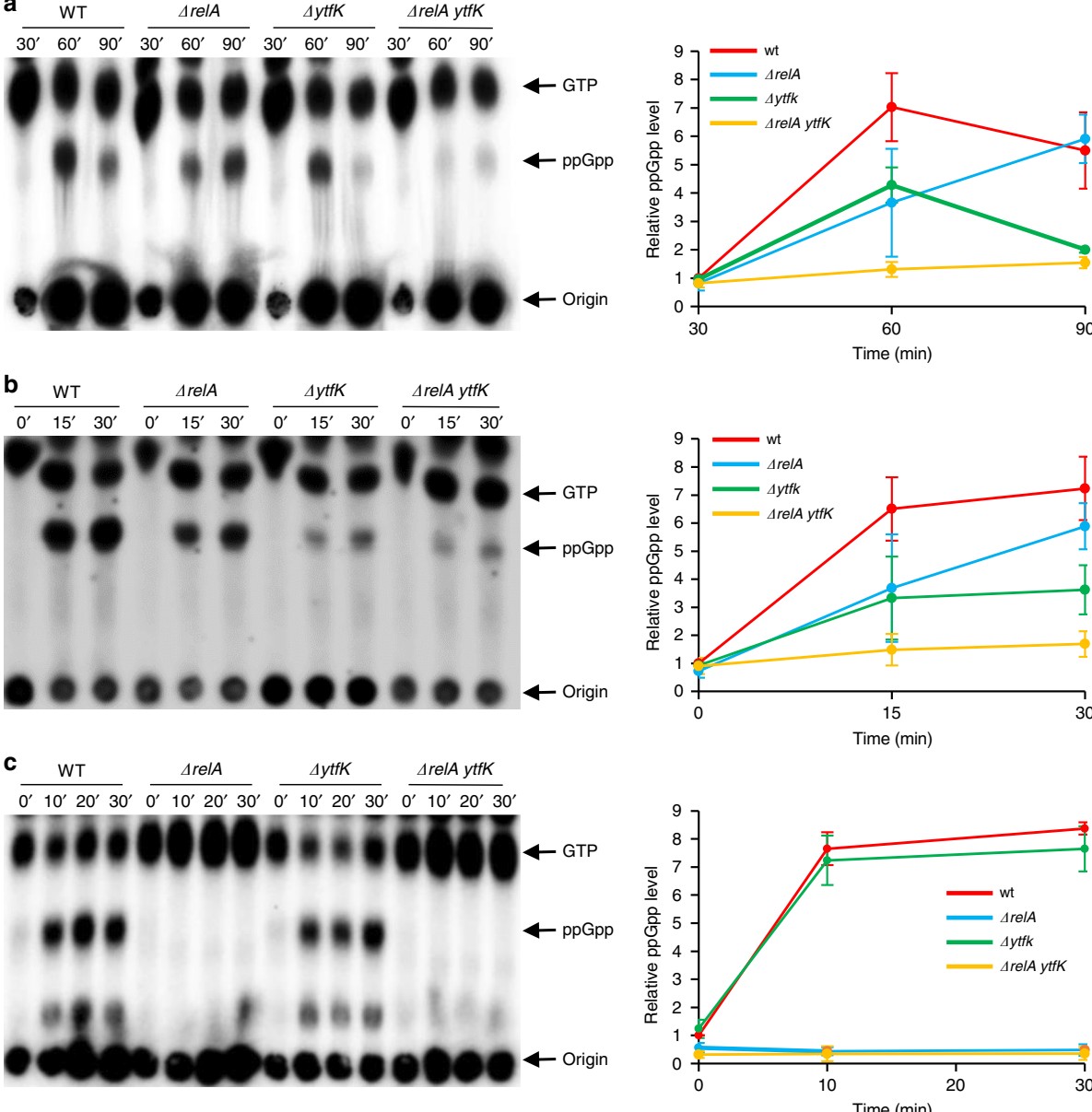

**Fig. 3 ytfK is required for normal accumulation of (p)ppGpp during phosphate and fatty acid starvation.** Exponentially growing cells of MG1655 (WT) and isogenic deletion strains, *ΔrelA*, *ΔytfK*, and *ΔrelA ytfK* were challenged for different starvations. Samples were taken at the indicated time, after being washed and resuspended in low-phosphate medium to induce phosphate starvation (**a**), after addition of 250 µg/mL cerulenin to induce fatty acid starvation (**b**), and after addition of 500 µg/mL L-Valine to induce amino acid starvation (**c**) (see the Methods section). Nucleotides were extracted and separated by TLC. Representative autoradiograph of the TLC is shown on the left, and quantification is presented on the right panel. Error bars indicate the standard deviations of averages of four independent experiments for phosphate starvation, six independent experiments for fatty acid starvation, and three independent experiments for amino acid starvation. Source data are provided as a Source Data file.

Pho regulon (PhoB) or one of the Pho products is involved in the control of (p)ppGpp accumulation during phosphate (Pi) starvation[28]. Since *ytfK* is a member of the Pho regulon[27,29] and is therefore induced during Pi starvation (Supplementary Fig. 3a), we naturally tested the hypothesis that YtfK could play a physiological role in (p)ppGpp accumulation under this condition. As shown in Fig. 3a, Pi starvation triggers fast accumulation of (p)ppGpp in wild-type (WT) *E. coli* strain. A similar accumulation is observed in a *ΔrelA* strain, consistent with the previous observation that *E. coli* cells starved for Pi accumulates (p)ppGpp in a SpoT-dependent manner[22,30]. Interestingly, deletion of *ytfK* reduces the accumulation of (p)ppGpp in the WT strain. Remarkably, this accumulation is abolished in the *ΔrelA ytfK*

strain. Taken together, our results show that YtfK is required to maintain normal (p)ppGpp level during Pi starvation (Fig. 3a).

The ability of SpoT to sense many sources of nutrient stress other than amino acid starvation has long been puzzling. However, an important mechanism allowing SpoT to sense FA synthesis limitation involving the acyl carrier protein (ACP) has been previously described[23,31,32]. We therefore tested the intriguing possibility that YtfK may also be required for (p)ppGpp synthesis under FA starvation. As shown in Fig. 3b, we observed a rapid and similar production of (p)ppGpp in cells of WT and the *ΔrelA* strain following FA starvation (induced by the addition of cerulenin, a specific inhibitor of FA synthesis), consistent with previous observations that FA starvation triggers

SpoT-dependent (p)ppGpp synthesis in presence of amino acids[20,23]. Importantly, deletion of *ytfK* impaired the accumulation of (p)ppGpp in response to FA starvation. Moreover, we observed that YtfK rapidly accumulated during FA starvation (Supplementary Fig. 3b). Taken together, these results point to an important role of YtfK in (p)ppGpp accumulation under this condition.

To further characterize the role of YtfK in (p)ppGpp physiology, we assessed its importance during amino acid starvation, a signal well known to trigger RelA-dependent (p)ppGpp synthesis[18]. To test this, isoleucine starvation was provoked by the addition of valine[33]. Under these conditions, we observed a rapid accumulation of (p)ppGpp in the WT strain. As expected, (p)ppGpp does not accumulate in the *ΔrelA* strain. Interestingly, we do not detect any differences in (p)ppGpp accumulation in the cells of a *ΔytfK* strain (Fig. 3c). Moreover, YtfK protein does not seem to accumulate after amino acid starvation (Supplementary Fig. 3c). These observations suggest that YtfK is not required to trigger RelA (p)ppGpp synthesis during amino acid starvation (Fig. 3c). Taken together, our results show that YtfK is essential for full accumulation of (p)ppGpp during stresses known to trigger SpoT-dependent synthesis of (p)ppGpp.

### YtfK enhances cell survival

Cerulenin is commonly considered as a bacteriostatic antibiotic. However, a recent study revealed that cerulenin becomes bactericidal concomitantly with a loss of cell envelope integrity in a strain unable to accumulate (p)ppGpp[34]. We therefore investigated the role of YtfK in cerulenin tolerance. As previously observed, the plating efficiency of WT cells was unaffected after 4 h of cerulenin treatment, while the plating efficiency of ppGpp[0] strain drops by more than 100,000-fold (Fig. 4a)[34]. Deletion of *relA* or *ytfK* does not significantly affect cerulenin tolerance (Fig. 4a). However, the plating efficiency of *ΔrelA ytfK* double-mutants drops by more than 10,000-fold after 4 h of cerulenin treatment, and this phenotype is trans-complemented by pEG220-*ytfK*. Interestingly, we also observed that while cerulenin efficiently inhibits cell growth, this inhibition is sharper in the WT and *ΔrelA* strains than what observed in the *ytfK* mutants and in the ppGpp[0] strain. These results suggest that (p)ppGpp level contributes to growth arrest during FA starvation (Figs. 3b, 4b; Supplementary Fig. 4a). Therefore, our results show that YtfK contributes to cerulenin tolerance, and further underscores the role of (p)ppGpp as a guardian of cell viability in the absence of FA synthesis[34].

Finally and consistent with the proposed role of (p)ppGpp as factor contributing to antibiotic tolerance[34–39], we observed that cerulenin pretreatment renders WT and *ΔrelA* mutant cells 10,000-fold more tolerant to ampicillin, but fails to substantially protect the *ΔytfK* mutant cells (Supplementary Fig. 4b).

### YtfK can interact with SpoT and modulate its activity

Several protein partners of the *E. coli* SpoT have been identified, and regulation of SpoT activities through protein–protein interactions have been reported[23,24,40]. We therefore tested whether YtfK is able to interact with SpoT in vivo using a bacterial two-hybrid (BTH) assay[41]. To this end, the complementary T18 and T25 domains of *Bordetella pertussis* adenylate cyclase were fused to N-terminus coding sequences of YtfK and SpoT and transformed in the *cya*-deficient *E. coli* strain BTH101. When this strain is co-transformed with both plasmids harboring T18 and T25 fusions, SpoT displayed a strong interaction in vivo with YtfK in both vector combinations. (Fig. 5a; Supplementary Fig. 5a).

Even though SpoT and RelA share clear sequence homologies and similarities, no interaction is detected between YtfK and full-length RelA by bacterial two-hybrid assay (Fig. 5a; Supplementary

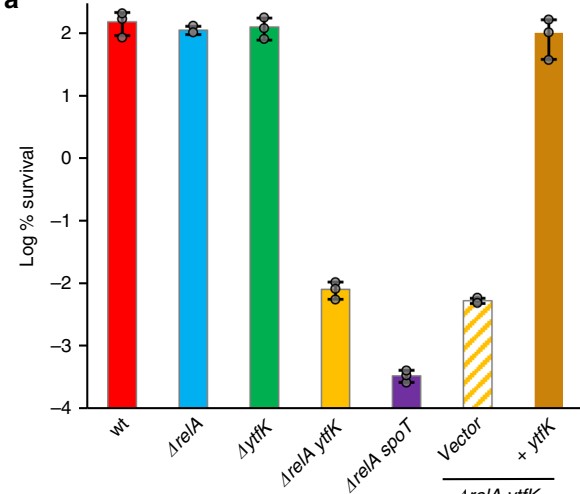

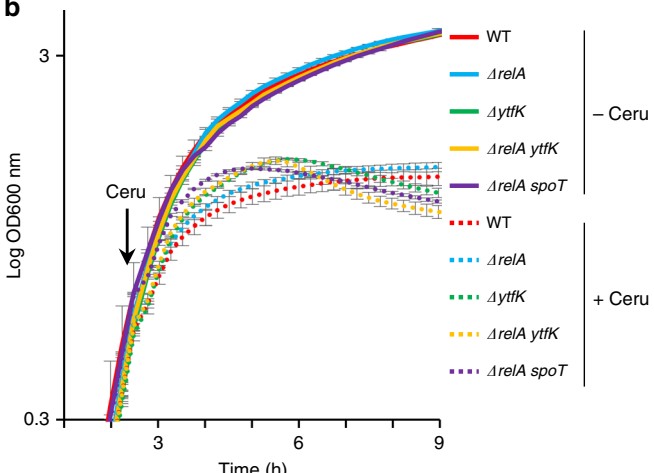

**Fig. 4 YtfK contributes to antibiotic tolerance. a** YtfK is required for cerulenin survival in absence of *relA*. Growing cells of MG1655 (WT) and isogenic deletion strains *ΔrelA*, *ΔytfK*, *ΔrelA ytfK*, *ΔrelA spoT*, and *ΔrelA ytfK* harboring the pEG220 and the *ΔrelA ytfK* harboring the pEG220-*ytfK* (with native promoter) were exposed to 500 μg/mL of cerulenin for 4 h (see the Methods section). Percentage of survival (log scale) after 4 h is shown. Error bars indicate the SDs of averages of three independent experiments. **b** Growth curve of WT and isogenic deletion strains *ΔrelA*, *ΔytfK*, *ΔrelA ytfK* and *ΔrelA spoT*, in presence (dashed line) or absence (solid line) of cerulenin. Overnight culture were 1000 times diluted in LB, and growth was monitored at 600 nm using a microplate reader. At the indicated time, 250 μg/ml of cerulenin was added. Error bars indicate the SDs of averages of three independent experiments. Source data are provided as a Source Data file.

Fig. 5b, c). To further gain insight into the YtfK-SpoT interaction and its role in (p)ppGpp synthesis, we randomly mutagenized *ytfK*. One mutant that had proline 42 substituted with leucine (P42L), exhibited impaired interaction with SpoT (Fig. 5a). Both T18-YtfK and T18-YtfK[P42L] recombinant proteins were correctly expressed as shown by western blot with Anti-CyaA (Supplementary Fig. 5d). Overexpression of *ytfK* but not *ytfK*[P42L] suppressed the growth defect of *ΔrelA* mutant on SMG plates (Supplementary Fig. 6a). This result predicts that the lack of interaction with SpoT abrogates the stimulatory effect of YtfK on (p)ppGpp accumulation. Indeed, it is the case. As judged by TLC analysis, *ytfK*[P42L] failed to promote (p)ppGpp synthesis when overexpressed in either WT or in *ΔrelA* strains in contrast to the

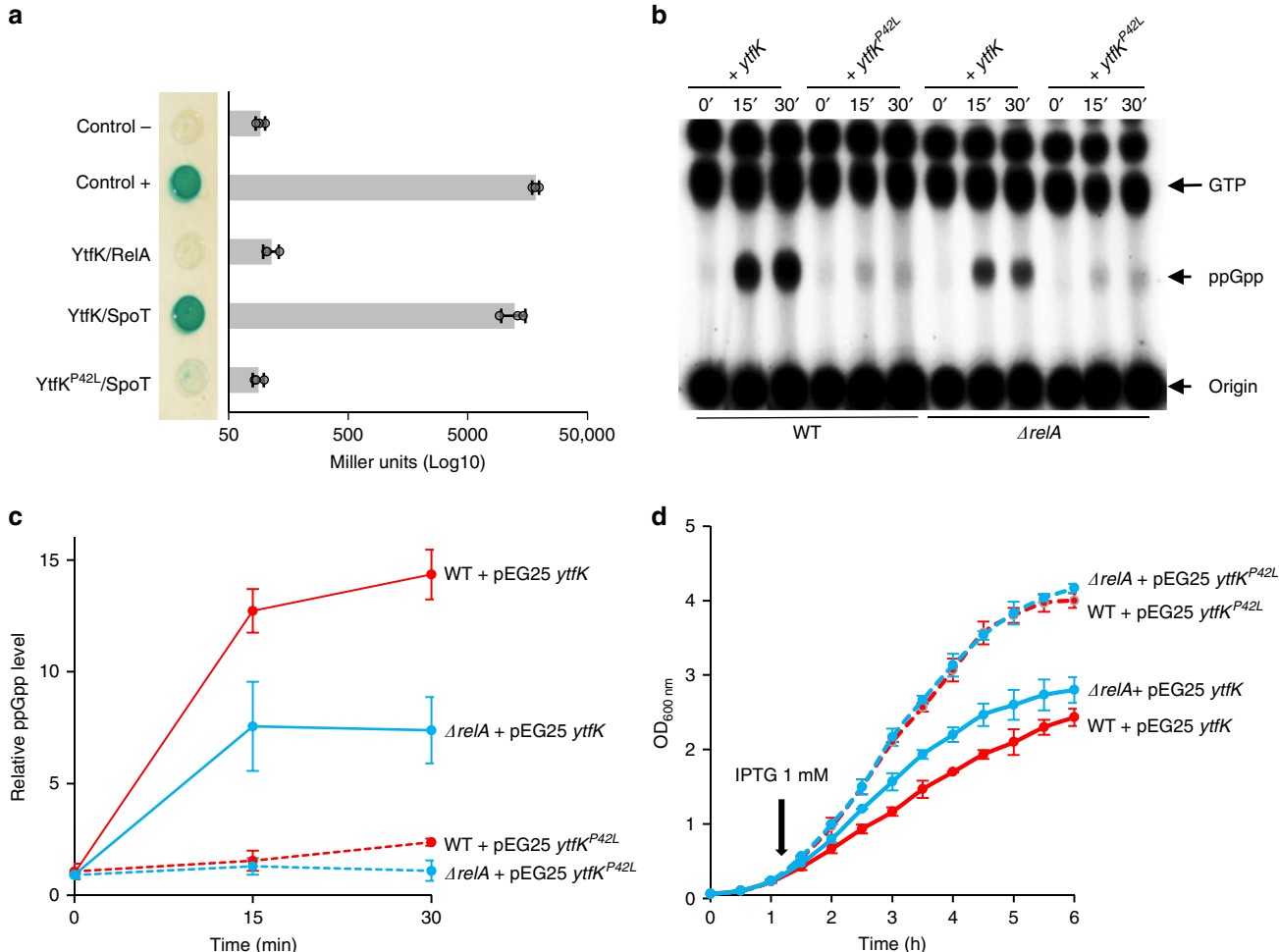

**Fig. 5 YtfK can interact with SpoT and trigger (p)ppGpp synthesis. a** YtfK interacts with SpoT in BTH assays. Briefly, overnight culture of BTH101 strain harboring pUT18c *ytfK* (or pUT18c *ytfK*$^{P42L}$) and pKT25-*spoT* (or pKT25 *relA*) were spotted on X-Gal agar base plates (see the Methods section). The blue color indicates a positive interaction. The results of the β-galactosidase assays using the same strains are shown on the horizontal graphs on the right. Error bars indicate the SDs of averages of three independent experiments. **b** In vivo (p)ppGpp accumulation following ectopic expression of both *ytfK* and *ytfK*$^{P42L}$. WT strain or *ΔrelA* mutant carrying *ytfK* or *ytfK*$^{P42L}$ on pEG25 were grown exponentially in phosphate MOPS minimal medium (see the Methods section). Samples were collected before and after *ytfK* induction (1 mM IPTG) prior to nucleotides extraction and were then separated by TLC. Representative autoradiograph of the TLC plates is shown and quantification is provided in (**c**). Error bars indicate the standard deviations of averages of three independent experiments. **d** The intracellular level of (p)ppGpp induced by *ytfK* overexpression controls the growth rate. Growth curve of WT (red) strain or *ΔrelA* mutant (blue) carrying *ytfK* (solid lane) or *ytfK*$^{P42L}$ (dashed lane) on pEG25 in the LB medium. Overnight culture was diluted 100 times, and growth was monitored at 600 nm. At the indicated time, 1 mM of IPTG was added to induce *ytfK* expression. Error bars indicate the standard deviations of averages of three independent experiments. Source data are provided as a Source Data file.

massive accumulation of (p)ppGpp observed when the WT copy of *ytfK* is induced (Fig. 5b, c).

Consistent with the proposed role of SpoT and (p)ppGpp in controlling the bacterial growth rate[42,43], we observed that induction of *ytfK* with 1 mM of IPTG strongly reduces the growth rate in both WT and *ΔrelA* strains, but not in the ppGpp⁰ strain (Fig. 5d; Supplementary Fig. 6b). Interestingly, the tougher growth rate control observed in WT strain is correlated to a higher (p)ppGpp level (Fig. 5c, d). Moreover, even if YtfK and YtfK$^{P42L}$ are produced at similar levels (Supplementary Fig. 6c, d), induction of *ytfK*$^{P42L}$ does not measurably affect the growth rate (Fig. 5d). Taken together, our results support the notion that a specific interaction between SpoT and YtfK controls SpoT activity, both during exponential growth (Fig. 5) and during nutrient stresses (Fig. 3).

**YtfK can interact with the catalytic domains of SpoT.** Using the BTH assay, we determined which domain(s) of SpoT are involved

in the interaction with YtfK. The RSH protein family has very similar domain structures with the N-terminal half of the protein comprising the synthetase (SYNTH) and hydrolase activities (HD) and the C-terminal part, involved in the regulation of these activities, encompassing a Thr-tRNA synthetase, GTPase, and SpoT domain (TGS domain); a helical domain; a domain containing conserved cysteines (CC domain); and an Aspartokinase, Chorismate mutase, and TyrA domain (ACT domain) (Fig. 6)[44]. To test which of these domains are required for YtfK interaction, we constructed several truncated SpoT proteins fused to the T25 domain of *B. pertussis* adenylate cyclase (Fig. 6).

The analysis revealed that YtfK interacts with the N-terminal half of SpoT comprising the catalytic domains (Fig. 6). Moreover, none of the five domains encompassing the C-terminal regulatory part of SpoT are required for this interaction. However, truncated SpoT proteins fusion lacking either the synthetase domain or the hydrolase domain fail to interact in vivo with YtfK (Fig. 6; Supplementary Fig. 7a, b). Thus, both catalytic domains seem to

be involved in the interaction with YtfK. Interestingly, the interaction with the N-terminal enzymatic half of SpoT seems specific, because YtfK does not interact with the homologous region of RelA by BTH assay (Supplementary Fig. 7c).

To confirm the SpoT-YtfK in vivo interaction revealed by BTH assay, we used BioLayer interferometry (BLI), an in vitro protein–protein interaction approach. SpoT catalytic domains (SpoT$^{1–378}$) and YtfK proteins were produced and purified by two consecutive affinity and size-exclusion chromatography experiments (Supplementary Fig. 8). YtfK was biotinylated and immobilized on streptavidin biosensors as the ligand. Interaction experiments were performed with purified SpoT$^{1–378}$ as the analyte. Upon addition of SpoT, a concentration-dependent association is recorded and decreases during the washing step of the sensor (dissociation step) (Fig. 7). These results indicate that YtfK interacts in vitro with the catalytic domains of SpoT with an estimated dissociation constant ($K_D$) of 35 μM (Fig. 7). Moreover, YtfK does not interact with the homologous N-terminal enzymatic half of RelA in vitro, supporting the specificity of interaction between YtfK and SpoT (Supplementary Fig. 9). Taken together, our results suggest that the catalytic domains of SpoT are necessary and sufficient to interact with YtfK.

## Discussion

In most gamma and beta-proteobacteria, to which E. coli belongs, the stringent response is orchestrated by two proteins RelA and SpoT encoding, respectively, for the (p)ppGpp synthase I and II activity. The metabolic cycle of (p)ppGpp is also further driven by the hydrolase activity encoded by SpoT, which is crucial for balancing the intracellular level of (p)ppGpp. Therefore, the molecular mechanisms regulating the switch between SpoT synthetase and hydrolase activities constitute a fine-tuning point of control for stress response and adaptation in E. coli. In this report, we discovered an important molecular mechanism regulating (p)ppGpp accumulation in E. coli. This regulation may involve physical interaction between the catalytic domains of SpoT and the protein YtfK. YtfK is a poorly characterized small basic protein specific to gamma-proteobacteria that seems limited to orders of Enterobacteriales, Vibrionales, Aeromonadales, Pasteurellales, and Alteromonadales[45]. We show that YtfK is important to maintain normal (p)ppGpp levels in response to both FA and phosphate starvation (Fig. 3). This regulation seems particularly important for cell survival. Indeed, we observed that ytfK actively contributes to cell viability under FA starvation (Fig. 4a). Moreover, it has been recently observed that deletion of ytfK also affects cell viability under phosphate starvation and $H_2O_2$ stress[46]. The stringent response is also recurrently linked to antibiotic tolerance and virulence. In this respect and consistent with the role of YtfK in controlling SpoT and (p)ppGpp levels, YtfK also contributes to stress-induced antibiotic tolerance (Supplementary Fig. 4b).

RelA and SpoT evolved from a common ancestor[9], however, despite the same domains organization and strong sequence homologies (30.5/64% identity/similarity), we were not able to

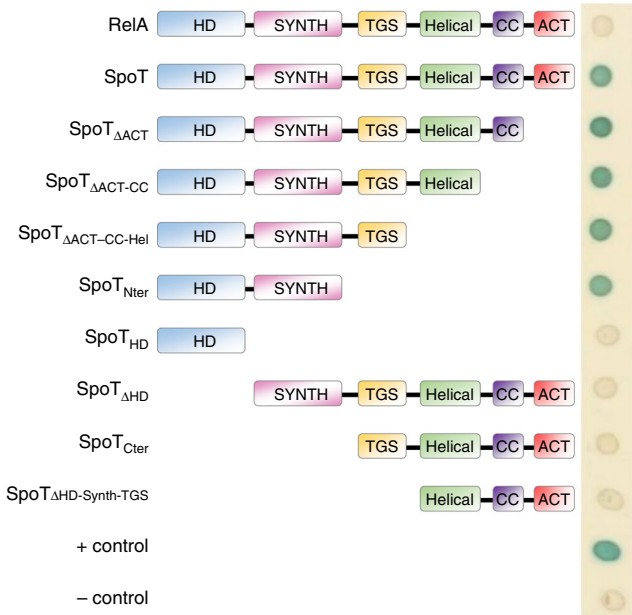

**Fig. 6 YtfK can interact with the catalytic domain of SpoT in vivo.** Bacterial two-hybrid assay with YtfK and several truncated version of SpoT. BTH101 strain harboring pUT18c ytfK and truncated pKT25 spoT were spotted on X-Gal agar base plates (see the Methods section). The blue color indicates a positive interaction. This experiment was repeated four times with identical results.

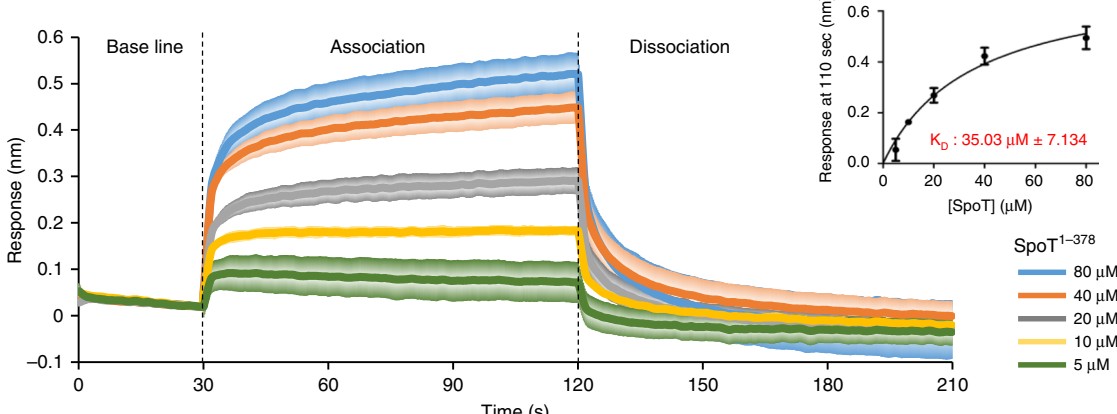

**Fig. 7 YtfK can interact with the catalytic domain of SpoT in vitro.** BioLayer interferometry assay. The graph shows the subtracted reference binding responses during the association and dissociation of SpoT on YtfK. For kinetic titration analysis: increasing concentrations of recombinant SpoT$^{1–378}$ (5–80 μM) were bounded to biotinylated YtfK (2.5 μM). Error bars (presented in degraded colors) indicate the standard deviations of averages of three experiments. The inset graph shows the specific BII response (nm) 10 s before the end of association as a function of SpoT$^{1–378}$ concentration (5–80 μM). Each data point (mean +/− SD) is the result from triplicates studies used to determine dissociation constant $K_D$. Source data are provided as a Source Data file.

detect physical interaction between YtfK and RelA (Fig. 5a; Supplementary Fig. 9), suggesting a specific physiological role of YtfK in triggering SpoT activities. The lack of functional interaction between RelA and YtfK is further supported by the observation that YtfK does not affect the level of (p)ppGpp under amino acid starvation, a signal well known to trigger (p)ppGpp synthesis by RelA[18] (Fig. 3c). However, we observed that YtfK triggers a stronger accumulation of (p)ppGpp when over-expressed in WT compared with the ΔrelA strain (Fig. 5b). Moreover, ectopic expression of ytfK reduced the growth rate of WT even more tightly to that observed in ΔrelA strain (Fig. 5d). Therefore, RelA also participates to the accumulation of (p)ppGpp promoted by YtfK. Even if we cannot rule out a direct activation of RelA by YtfK under these conditions, we suggest that this effect is rather indirect by residual activation of RelA. Indeed, it is well described that the hydrolysis activity of SpoT is required for balancing (p)ppGpp level in the presence of RelA, and disruption of the spoT gene in E. coli is therefore lethal[12]. Moreover, it is currently admitted that RSHs avoid to simultaneously synthesize and degrade (p)ppGpp primarily to prevent futile cycle[10]. Thus, we propose that YtfK pushes the catalytic balance of SpoT toward (p)ppGpp synthesis rather than hydrolysis therefore enabling residual activation of RelA to maximize the alarmone production. Therefore, the regulation of SpoT activities by YtfK points to an additional layer of regulation to the current stringent response model.

SpoT functions as a central protein that integrates various nutritional stress signals[12,20–22]. How different types of nutrient starvation can use the same or independent signal transduction pathways to trigger SpoT-dependent (p)ppGpp accumulation remains still unclear. It has been shown that the ACP transduces the status of FA metabolism to SpoT, thereby triggering (p)ppGpp accumulation in response to FA starvation[23,31,32]. Moreover, FA metabolism could also be the transducing signal that triggers SpoT during carbon starvation. Indeed, carbon depletion would lead to FA starvation through a decrease of the acetyl-CoA pool that is supplied by glycolysis[47]. Our finding that YtfK, in addition to the well-described role of ACP, is required for (p)ppGpp accumulation under FA starvation renders more sophisticated the current model for the switch linking SpoT to fatty acid metabolism[23,31,32]. Interestingly, YtfK protein level plays an important role for adjusting intracellular (p)ppGpp level. Indeed, ectopic expression of ytfK is sufficient to trigger SpoT-dependent (p)ppGpp accumulation in absence of nutritional stresses (Figs. 1c, 2). Moreover, ytfK is rapidly expressed under Pi and FA starvation, and is required for full (p)ppGpp accumulation under these conditions (Fig. 3a, b; Supplementary Fig. 3a, b) strongly arguing that YtfK protein level is indeed important to trigger the stringent response under nutrient stress conditions. However, how the control of YtfK protein level becomes critical for adjusting (p)ppGpp level in response to other stresses known to trigger SpoT-dependent (p)ppGpp accumulation remains unknown. Interestingly, ytfK is also induced and involved in H$_2$O$_2$ tolerance by stimulating expression of at least the detoxifying enzyme encoded by katG[46]. Whether this stimulation is direct or indirect through activation of SpoT remains to be uncovered.

The CTD region of RSH enzymes are thought to play pivotal roles in sensing nutrient starvation and to controlling the reciprocal regulation of the two enzymatic states[10,48,49]. Consistent with a regulatory role of the CTD, the TGS domain is required for ACP and Rsd binding to SpoT of E. coli. Moreover, the ACT domain is also needed for interaction with the phosphorylated enzyme IIA$^{Ntr}$ to promote (p)ppGpp accumulation in Caulobacter crescentus in response to glutamine deprivation[50,51]. In contrast, we observed that the NTD region of SpoT is necessary

and sufficient for YtfK binding (Figs. 6, 7), suggesting a distinct mode of regulation. The crystal structure of the HD and SYNTH domains of the Rel/Spo homologue from Streptococcus equisimilis has revealed two conformations of the enzyme corresponding to reciprocal activity states: (p)ppGpp-hydrolase-OFF/synthetase-ON and (p)ppGpp-hydrolase-ON/synthetase-OFF[11]. A docking experiment using modeled structures predicts that YtfK might bind SpoT at the small synthetase/hydrolase interdomain interface (Supplementary Fig. 10). The molecular basis of the apparent regulation of SpoT by YtfK can only be speculated upon. However, based on our preliminary in vitro results (Fig. 7), we propose that YtfK might directly interact with SpoT to trigger conformational changes in the catalytic domains. Alternatively, additional indirect effects of YtfK on SpoT activity are also possible, and further structural and functional analysis are currently under investigation to test whether direct binding of YtfK at the interface between the opposing site of SpoT affects the antagonistic regulation of (p)ppGpp synthesis and degradation.

## Methods

**Bacterial strains and plasmids**. Bacterial strains and plasmids are listed in Supplementary Table 1 and Supplementary Data 1, respectively. DNA oligonucleotides are listed in Supplementary Data 2.

**MG1655 YtfK::mNeonGreen construction**. The chromosomal YtfK::mNeon-Green fusion was generated by lambda red recombination adapted from Blank et al.[52]. The DNA fragment was amplified by PCR using primers pEJM230/231 with plasmid pWRG100 as a template[52]. The resulting PCR product was electroporated into MG1655 cells after 1 h of lambda recombinase expression from pKD46 plasmid[53]. After 1 h of phenotypic expression, the cells were plated on nutrient agar (NA) containing 25 μg/mL chloramphenicol. Selected colonies contained a ytfK::sce-I::cm cassette which has been then newly transduced in MG1655 cells harboring the pWRG99 plasmid[52]. The chloramphenicol cassette was then removed by counter selection using lambda red recombination to insert a PCR product complementary to the flanking regions of the sce-I::cm cassette on the chromosome. The PCR product was generated with primers pEJM244/245 on pJF119EH encoding mNeonGreen as a template (Generous gift from Axel Magalon's lab). PCR product was then electroporated into MG1655 ytfK::sce-I::cm expressing lambda recombinase from pWRG99[52], and after 1 h of phenotypic expression serially diluted and plated on NA containing 100 μg/mL ampicillin and 1 μg/mL anhydrotetracycline. pWRG99 also encoded for the meganuclease I-Sce-I under the control of an anhydrotetracycline (aTc) inducible promoter. The proper integration of the mNeonGreen was confirmed by diagnostic PCR and then sequenced.

**Plasmid construction**. pEG25 relA and spoT: Due to their intrinsic activities, RelA and SpoT overproduction become toxic. To circumvent this problem and therefore be able to express relA in ΔrelA spoT strain, we have engineered a robust system to tightly regulate intracellular level of RelA. First, we have constructed the pEG25 plasmid which contains the Pt5 promoter from the pCA24N. This promoter contains three Lac operators, and is therefore less leaky (see below). Most importantly, we played with constants that contribute with the process of translation initiation. Indeed by engineering the Shine-Dalgarno sequence, the initiation codon and the spacer between them we were able to tightly regulate the intracellular level of RelA (SD4 TTG) and SpoT (SD8 TTG). The chosen Shine-Dalgarno sequences are described in the primers list.

pEG25: pMG25 derivative where the P$_{Lac}$ promoter has been replaced by P$_{t5}$ from the pCA24N[26]. P$_{t5}$ promoter has been amplified by PCR using primers EG186/EG187, digested with XhoI and EcoRI, and ligated in the plasmid digested with the same enzymes.

pEG25 derivatives: Genes inserted have been PCR amplified from the chromosomal DNA and with primers listed in the Supplementary Data 2, digested with EcoRI and BamHI and ligated in the pEG25 digested with the same enzymes.

pBbS2K: The original plasmid pBbS2K(RFP) has been digested with BglII/BamHI, and then self ligated. The resulting plasmid has been then used as a negative control.

pBbS2K(RFP) derivatives plasmids: Genes inserted have been PCR amplified from the chromosomal DNA with primers listed in the Supplementary Data 2, digested with EcoRI/BamHI, and ligated in the pBbS2K(RFP) plasmid digested with the same enzymes.

pEG220: pNDM220 (Amp) has been engineered and modified to a kanamycin plasmid. Kanamycin cassette has been amplified from pKD4 using pEG287/288 and transformed to DY331 strain with the lambda red recombineering system on the chromosome. Antibiotic cassette exchange has been checked by PCR and then sequenced.

pUT18C-derivative plasmids: Genes inserted have been PCR amplified from the chromosomal DNA (or mutated plasmid) with primers listed in the Supplementary Data 2, digested with XbaI and KpnI, and ligated in pUT18C digested with the same enzymes.

pKT25-derivative plasmids: Genes inserted have been PCR amplified from the chromosomal DNA (or from mutated plasmid) with primers listed in Supplementary Data 2, digested with XbaI and KpnI, and ligated in pKT25 digested with the same enzymes.

pET28a + -TRX: TRX sequence, including *thioredoxin-6his-tev*, has been amplified from pLic07 with primers listed in Supplementary Data 2, digested with NcoI and BamHI, and ligated in pET28a + digested with the same enzymes.

pET28a + -*ytfK*: *ytfK* gene has been amplified from *E. coli* chromosomal DNA with primers listed in Supplementary Data 2, digested with BamHI and HindIII, and ligated in pET28a + -TRX digested with the same enzymes.

**Media and antibiotics.** Luria—Bertani broth from OXOID (LP0021B; LP0042B) was prepared as previously described[54]. MOPS minimal medium was prepared as previously described[55], and was supplemented with 2 mM phosphate (or 0.4 mM phosphate for uniform $P^{32}$ labeling), 0.2% glucose, and 40 μg/ml (each) of the 20 amino acids. Terrific broth (TB) :1.2% peptone, 2.4% yeast extract, 72 mM $K_2HPO_4$, 17 mM $KH_2PO_4$, and 0.4% glycerol.

Rich solid medium used in this study is NA from Oxoid (CM0003B). M9 minimal medium: M9 salt (60 mM $Na_2HPO_4$; 22 mM $KH_2PO_4$; 8 mM NaCl; 20 mM $NH_4Cl$) 1 mM $MgSO_4$; 100 μM $CaCl_2$; thiamine 1 μg/mL; 0.2% glucose. SMG solid minimal medium: M9 medium supplemented with 40 μg/mL of L-serine; 40 μg/mL L-methionine; 40 μg/mL glycine; and 15 g/l of agar.

When required, media was supplemented with antibiotic (50 μg/ml ampicillin; 50 μg/ml chloramphenicol; 25 μg/ml kanamycin). Expression of protein from plasmids carrying $P_{Lac}$ promoter was induced by addition of IPTG.

**Suppression of the nongrowing phenotype of Δ*relA* on SMG.** A pooling mixture of the 6His-tagged genes (minus GFP) ASKA plasmid library[26] has been transformed into a Δ*relA* deletion strain. Cell were then spread on SMG plates supplemented with chloramphenicol and different concentration of IPTG (0, 100, and 200 μM) to induce the expression of ASKA plasmid-encoded genes and incubated for 36 h at 37 °C. Plasmids that produced colonies under this nonpermissive condition were sequenced, and we identified *ytfK*, *metL*, *sdaB*, *ymgB*, *yaaA*, *thiI*, *murI*, and *relA* as candidate genes that enable growth in a Δ*relA* strain on high concentration of serine, methionine, and glycine (SMG)[25].

**Bacterial two-hybrid assay (BTH).** We used BTH assay to test protein–protein interaction in vivo. This method is based on the reconstitution of the adenylate cyclase from *Bordetella pertussis*[41]. Proteins of interest were fused to T18 and T25 fragments of adenylate cyclase using pUT18c and pKT25. After co transformation in BTH101 strain, the plates were incubated overnight at 30 °C. Colonies were re-streaked overnight at 30 °C. Then, 1 mL of LB supplemented with ampicillin and kanamycin was inoculated in the morning and incubated at 30 °C with shaking for 8 h. In total, 5 μl of undiluted cultures were then spotted on NA agar plates supplemented with X-Gal 40 μg/mL with different concentrations of IPTG. For β-galactosidase experiments, the same inoculum has been used, and β-galactosidase activity was determined as described by Miller with the use of the TECAN microplate reader to follow $OD_{600}$ and $OD_{420}$ over time.

**Screening for loss of interaction by *ytfK* random mutagenesis.** pUT18c-*ytfK* plasmid has been mutagenized randomly through propagation in the mutator XL1-Red strain from Agilent Technologies. XL1-Red strain are deficient in three of the primary DNA repair pathways (*mutS*, *mutD*, and *mutT*). BTH101 harboring pKT25-*spoT* was then transformed with the resulting pUT18c-*ytfK* mutated plasmid library and screened on X-Gal plates. White colonies were selected and sequenced.

**In vivo (p)ppGpp measurement.** (p)ppGpp accumulation was measured in cells grown and uniformly labeled with $^{32}$P in MOPS minimal medium supplemented as described below. In total, 100 μl samples were taken at the indicated times, and 40 μl of 21 M formic acid was added to stop the reaction. Samples were then left on ice for 20 min, and spun down at 14,000 *g* for 20 min at 4 °C. In all, 5 μl of each sample were loaded on PEI Cellulose TLC plates (purchased from MerckK-Millipore). Plates were revealed by PhosphoImaging (GE Healthcare) and analyzed using ImageQuant software (GE Healthcare). Quantification was performed using FIJI software. ppGpp levels were normalized to that of the WT strain at time zero for each starvation condition.

For the phosphate starvation, overnight cultures were grown in MOPS 2 mM phosphate supplemented with 40 μg/mL of the 20 amino acids and 0.2% glucose. Overnight cultures were then diluted 1/100-fold in the same medium and incubated at 37 °C with shaking up to $OD_{600}$: 0.4–0.5. Cells were harvested and washed once in MOPS without phosphate and then resuspended in MOPS with 50 μM phosphate and labeled with 150 μCi of $^{32}$P.

For fatty acid starvation, overnight cultures were grown in MOPS 2 mM phosphate supplemented with 40 μg/mL of the 20 amino acids and glucose 0.2%.

Overnight cultures were then diluted 1/100-fold in the same medium with 0.4 mM phosphate instead of 2 mM and incubated at 37 °C with shaking up to $OD_{600}$: 0.4–0.5. Cells were then diluted down to $OD_{600}$: 0.05 in the same medium, labeled with 150 μCi of $^{32}$P and grew up to $OD_{600}$: 0.25. Fatty acid starvation is then induced by addition of 250 μg/mL of cerulenin.

For amino acid starvation, cells were grown in MOPS 2 mM phosphate supplemented with 0.2% glucose without amino acids. Overnight cultures were then diluted 1/100-fold in the same medium with 0.4 mM phosphate instead of 2 mM and incubated at 37 °C with shaking up to $OD_{600}$: 0.4–0.5. Cells were then diluted down to $OD_{600}$: 0.05 in the same medium, labeled with 150 μCi of $^{32}$P and grew up to $OD_{600}$: 0.25. Amino acid starvation is then induced by addition of 500 μg/mL of L-Valine.

For *yfK* overexpression, WT strain and Δ*relA* mutant harboring pEG25 *ytfK* or pEG25 *ytfK-6his* and pEG25 *ytfK^{P42L}* or pEG25 *ytfK^{P42L}-6his* were grown in MOPS 2 mM phosphate supplemented with 40 μg/mL of the 20 amino acids and glucose 0.2%. Overnight cultures were then diluted 1/100-fold in MOPS 0.4 mM phosphate and incubated at 37 °C with shaking up to $OD_{600}$: 0.4–0.5. Cells were then diluted down to $OD_{600}$: 0.05 in the same medium, labeled with 150 μCi of $^{32}$P and grew up to $OD_{600}$: 0.25. *ytfK* expression was then induced by the addition of 1 mM of IPTG.

**Cerulenin antibiotic tolerance.** Strains of WT, Δ*relA*, Δ*ytfK*, Δ*relA ytfK*, Δ*relA spoT*, Δ*relA ytfK* harboring the pEG_{220} and the Δ*relA ytfK* harboring the pEG_{220}-*ytfK* were tested for the effect of 500 μg/mL of cerulenin. Overnight cultures were diluted 1000-fold in 10 ml of fresh LB medium, and incubated ~2.5 h at 37 °C with shaking (typically reaching $1–2 \times 10^8$ cells/mL). For complementation assay, the medium was supplemented by 25 μg/ml of kanamycin. Then aliquots of 1 mL were transferred into a 15 mL falcon tube containing 10 μL of cerulenin 50 mg/mL and incubated 4 h at 37 °C with shaking. For determination of CFUs, 200 μl aliquots were removed, the cells harvested, resuspended in fresh medium, serially diluted, and plated on solid NA medium. Surviving fraction was calculated by dividing the number of CFU/ml in the culture after 4 h of incubation with the cerulenin by the number of CFU/ml in the culture before adding the cerulenin.

**Ampicillin tolerance after cerulenin pre-treatments.** WT, Δ*relA*, Δ*ytfK*, Δ*relA ytfK*, and Δ*ytfK* harboring the pEG_{220} and the Δ*ytfK* harboring the pEG_{220}-*ytfK* were challenged against ampicillin treatment (100 μg/ml) after being pre-treated with 250 μg/ml of cerulenin to induce (p)ppGpp synthesis in a SpoT-dependent manner. Overnight cultures were diluted 1000-fold in 10 ml of fresh LB medium and incubated for about 2.5 h at 37 °C with shaking (typically reaching $1–2 \times 10^8$ cells/mL). For complementation assay, medium was supplemented with 25 μg/ml of kanamycin. Then 3 mL aliquots were transferred into a 50 -mL falcon tube containing 15 μL of cerulenin 50 mg/mL and incubated during 30 min at 37 °C with shaking and then treated for 4 h with ampicillin (100 μg/ml). For determination of CFUs at the indicated time points, 800 μl aliquots were removed, the cells harvested, resuspended in fresh medium, serially diluted, and plated on solid NA medium. Surviving fraction was calculated by dividing the number of CFU/ml in the culture after 4 h of incubation with ampicillin by the number of CFU/ml in the culture before adding ampicillin.

**Protein production.** For SpoT$^{1–378}$ and RelA$^{1–396}$ (catalytic domains) production, BL21(DE3) competent cells were transformed with the pMG25-SpoT$^{1–378}$ -6His or pEG25-RelA$^{1–396}$ -6His and plated on NA plates containing 100 mg/L ampicillin. Several colonies were picked up and inoculated into 100 ml of LB containing 100 mg/L ampicillin. The cultures were grown with shaking at 30 °C overnight. Overnight cultures were dispersed as inoculum of 20 mL aliquots into 1L of TB media with of 100 mg/L of ampicillin and shaken at 30 °C until $OD_{600}$ reached 0.5, then cells were induced with 0.5 mM IPTG and incubated at 30 °C for 4 h.

For YtfK production, BL21 (DE3) competent cells were transformed by the pET28a-*ytfK* plasmid and plated on NA plates with 50 mg/L kanamycin. Several colonies were picked up and inoculated into 100 ml LB containing 50 mg/L kanamycin. The cultures were grown with shaking at 30 °C overnight. Overnight cultures were dispersed as inoculum of 20 ml aliquots into 1L LB media in the presence of 50 mg/L of kanamycin and shaken at 30 °C until $OD_{600}$ 0.4 unit, then cells were induced with 0.5 mM IPTG and incubated at 20 °C for 4 h.

Finally, cells were centrifuged at $9000 \times g$ for 20 min at 4 °C. Dry cell pellet was stored at −80 °C.

**Protein purification.** For YtfK purification, cells were harvested from 4L culture, suspended in lysis buffer (Tris-HCl 50 mM pH 8.0, NaCl 300 mM, EDTA 1 mM, lysozyme 0.5 mg/mL, phenylmethylsulfonyl fluoride (PMSF)), DNase 20 μg/mL, and $MgCl_2$ 20 mM. The mixture was incubated for 1 h at 4 °C with gentle shacking and then subjected to two cycles of French-press lysis steps. The soluble fraction was obtained by centrifugation for 30 min at $200,000 \times g$. Recombinant proteins were purified by ion metal affinity chromatography using a 5-mL Nickel (HiTrap$^{HP}$) Column on an ÄKTA pure 25 (GE healthcare) pre-equilibrated in Tris-HCl 50 mM pH 8.0, NaCl 300 mM, 10 mM imidazole (buffer A). After several washes in buffer A, 6 × His-tagged proteins were eluted in buffer A supplemented with 250 mM imidazole and immediately desalted using Hiprep 26/10 Desalting column pre-equilibrated with buffer A. The resulting desalted proteins were mixed

with 0.2 mg/mL of TEV protease and incubated for 2 h at RT and then loaded onto a HisTrap™ column pre-equilibrated in buffer A, which selectively retains the TEV, TRX, and the uncleaved proteins and contaminants. Untagged YtfK was collected in the flow through, concentrated on a Centricon (Millipore; cutoff of 3 kDa), and passed through a HiLoad 16/600 Superdex 200 column pre-equilibrated with 50 mM Tris-HCl pH 8.0, 500 mM NaCl, 500 mM KCl, 2 mM β-mercaptoethanol, glycerol 2%. The purity of YtfK preparations was assessed by SDS–PAGE (Supplementary Fig. 8a, b) and spectrophotometrically [OD260/OD280] ratio below 0.52.

For SpoT$^{1-378}$ and RelA$^{1-396}$ purification, cells were harvested from 4L culture, then cell pellets were resuspended in lysis buffer composed of 50 mM Tris-HCl pH 8.0, 500 mM NaCl, 10 mM imidazole, 2 mM β-mercaptoethanol, 0.5% CHAPS, glycerol 2%, 1 mM EDTA, 0.5 mg/mL lysozyme, 1 mM phenylmethylsulfonyl fluoride, 20 μg/mL DNase, and 20 mM MgCl₂. The mixture was incubated for 1 h at 4 °C with gentle shaking. Further French-Press lysis step was performed to ensure the complete cell lysis. Pellet and soluble fractions were separated by centrifugation for 30 min at 20,000 × g. The soluble fraction containing SpoT$^{1-378}$ or RelA$^{1-396}$ proteins were loaded at room temperature onto a 5-ml nickel column (HiTrap$^{HP}$) using an ÄKTA pure 25 apparatus (GE healthcare) pre-equilibrated with equilibrium buffer 50 mM Tris-HCl pH 8.0, 500 mM NaCl, 10 mM imidazole, 2 mM β-mercaptoethanol, glycerol 2%, and the immobilized proteins were eluted in elution buffer 50 mM Tris-HCl pH 8.0, 500 mM NaCl, 500 mM imidazole, 2 mM β-mercaptoethanol, and glycerol 2%. Eluted proteins were immediately subjected to size-exclusion chromatography (SEC) purification using a HiLoad 26/600 Superdex 200 pg column pre-equilibrated with 50 mM Tris-HCl pH 8.0, 500 mM NaCl, 500 mM KCl, 2 mM β-mercaptoethanol, and glycerol 2%.

The purity of SpoT and RelA preparations was assessed by SDS–PAGE (Supplementary Fig. 8c–e), and spectrophotometrically [OD260/OD280] ratio below 0.61 for SpoT and below 0.57 for RelA.

**Biolayer interferometry (BLI)**. YtfK was biotinylated using the EZ-Link NHS-PEG4-Biotin kit (Perbio Science, France) with 1:1 ratio of YtfK and Biotin at 4 °C, respectively. The reaction was stopped, by removing the excess of the biotin using a Zeba Spin Desalting column (Perbio Science, France). BLI studies were performed at 25 °C using the Blitz apparatus (ForteBio, USA) with shaking at 2200 rpm with the following steps; 30 s baseline, 90 s association, and 90 s dissociation. Streptavidin biosensor tips (ForteBio, USA) were first hydrated in 0.2 ml of 50 mM Tris-HCl pH 8.0, 500 mM NaCl, 500 mM KCl, 2 mM β-mercaptoethanol, glycerol 2% for 10 min, and then loaded with 2.5 μM of biotinylated YtfK in the same buffer. To study the binding of YtfK to SpoT$^{1-378}$ or RelA$^{1-396}$, increasing concentrations of SpoT$^{1-378}$ (5 to 80 μM) or RelA (40 to 120 μM) in interaction buffer (50 mM Tris-HCl pH 8.0, 500 mM NaCl, 500 mM KCl, 2 mM β-mercaptoethanol, glycerol 2%, and BSA 1 mg/ml) were used, and the association and dissociation phases were monitored, respectively, with constant agitation at 2200 rpm. To avoid the non-specific binding of SpoT$^{1-378}$ or RelA$^{1-396}$ to the Streptavidin biosensors, the biosensors were incubated with 10 μg/mL biocytin in interaction buffer (50 mM Tris-HCl pH 8.0, 500 mM NaCl, 500 mM KCl, 2 mM β-mercaptoethanol, glycerol 2% BSA 1 mg/ml) for 90 s. In all experiments, the BLItz Pro™ software performed a reference subtraction of the SpoT$^{1-378}$ (or RelA$^{1-396}$) protein response on the uncoated biosensors for each tested concentrations. The dissociation constants (K$_D$), i.e., affinity of YtfK for SpoT, were estimated using the GraphPad Prism 5.0 software on the basis of the maximum responses in nm at 110 s, directly related to the concentration of SpoT$^{1-378}$. The K$_D$ was estimated by plotting on the x-axis, the different concentration of SpoT$^{1-378}$ and the different responses of SpoT$^{1-378}$ (10 s before the end of the association) on the y-axis. For K$_D$ calculation, a non-linear regression fit for xy analysis was used with a one site (specific binding) as a model which corresponds to the equation $y = Bmax \times x/(K_D + x)$.

**Western blot analysis**. The western blot using Penta-His-HRP-conjugated against 6His-tagged proteins has been done following the recommendation of the company (Qiagen ref: 34460). Cells were grown exponentially at 37 °C in LB (OD$_{600}$ 0.5), then cells were induced with 200 μM of IPTG for 1 h. Sample was prepared, and equal amount of total proteins were loaded per lane on SDS–PAGE. Immunoblot analysis was performed with the antibody diluted 1/10,000 and visualized by using luminata crescendo HRP substrate (Millipore ref: WBLUR0100) and ImageQuant Las4000 (GE Healthcare).

The western blot experiment using anti-cyaA antibody against adenylate cyclase protein encoded by pUT18c plasmid has been performed using PBS + 0.1% Tween20 (PBST) for all steps of the immunoblot. The membrane was incubated in blocking buffer for 1 h (PBST + 5% milk). The first anti-cyaA antibody has been diluted 1/5.000 (Santa Cruz, ref: sc-13582) in PBST milk and incubated for 1 h at room temperaure. The secondary antibody (Sigma, ref: A4416) has been diluted 1/10.000 in PBST milk and incubated for 1 h at room temperature. Immunoblot was visualized by using luminata crescendo HRP substrate (Millipore, ref: WBLUR0100) and ImageQuant Las4000 (GE Healthcare). For sampling, MG1655 was transformed with the all different truncated versions of SpoT encoded by pUT18c. Cells grew during 3H00 (OD: 0.5) at 30 °C in LB. Protein productions were induced by addition of 1 mM IPTG for 4H00. Sample was prepared and equal amount of total proteins were loaded per lane on SDS–PAGE.

**Protein modeling and molecular docking simulation**. The available atomic coordinates of RelA (PDB ID: 5KPV) from *E. coli* was used as a template to generate a model of *E. coli* SpoT (30.4%/64.3% identity/similarity) using SWISS-MODEL (a web-based integrated service for protein structure homology modeling)[56]. Similarly, the model of *E. coli* YtfK was generated using the NMR structure of the YtfK homologue from *Shewanella oneidensis* (PDB ID: 2JRO; 30.3%/65.8% identity/similarity with *E. coli* YtfK).

For the modeling of YtfK-SpoT$^{1-378}$ binding interface, we used the HADDOCK2.2 webserver interface. To define Haddock run restraints, we used the surface exposed residues of both YtfK and SpoT as active residues that could be directly involved in the interaction. For SpoT and YtfK, we selected the surface exposed residues manually using PyMol (Supplementary Table 2). Three independent docking simulations were performed. For each run, ten clusters were generated, and classified based on their HADDOCK score. Five YtfK/SpoT models picked from the best five clusters are shown in Supplementary Fig. 10, and the parameters of the respective clusters generated by Haddock are shown in Supplementary Table 3.

**Fluorescence microscopy and analysis**. Overnight culture of WT and *ytfK*-L-mNeonGreen strains in MOPS minimal medium were diluted 1000 times in the same medium and incubated at 37 °C until OD$_{600}$ nm reached 0.25. Cells were then starved using the same protocol used to measure (p)ppGpp level in vivo. The microscopy images were taken using an inverted epifluorescence microscope Nikon Eclipse TiE PFS (100 × oil objective NA 1.45 Phase Contrast) and a Hamamatsu Digital camera C11440 ORCA-Flash4.0 LT. Images were collected with NS elements software. Samples (3 μl of cells) were spotted on a 24 × 50 mm coverslip, and covered directly with a thin pad of minimal medium supplemented with 1.5% agarose. Twenty fields were taken for each point using the perfect focus system (PFS) and a motorized stage. The fluorescence images have been acquired with diascopy (DIA) and a GFP filter (482/35 nm excitation, 536/40 nm emission). The exposure time were 30 ms and 300 ms, respectively. The images were analyzed in TIFF format with the plugin MicrobeJ[57] available for downloading in FIJI software.

A threshold was determined for each experiment to detect bacteria. Then, the backgrounds were subtracted with a rolling ball radius of 30 pixels, and the fluorescence of each conditions was measured with MicrobeJ. All the images have the same contrast and have been acquired and analyzed in the same way.

**Reporting summary**. Further information on research design is available in the Nature Research Reporting Summary linked to this article.

## Data availability

All relevant data are available in this article and its Supplementary Information files. The source data underlying Figs. 1c, 3a–c, 4a, b, 5b–d and 7, and Supplementary Figs. 4b, 5d, 6b–d, 7b, and 9 are provided as a Source Data file. Any additional data relevant to this paper are available from the corresponding authors upon request.

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

## Acknowledgements

This work was supported by the European Research Council starting grant (ERC StG) under the European Union's Horizon 2020 and innovation program grant agreement no. 714934 "Stringency" to E.M. We thanks Axel Magalon for the generous gift of the pJF119EH encoding for the mNeonGreen. We thank Mike Cashel, Aurelia Battesti, Anne Galinier, and the members of the Maisonneuve group for stimulating discussions.

## Author contributions

E.M. and E.G. conceptualized the study. E.M., E.G., P.G., D.B., B.D. and M.D. performed the experiments. E.M., E.G., P.G., D.B., B.D. and M.D. analyzed the data. E.M. and E.G. wrote the original draft of the paper. D.B., reviewed and edited the paper. E.M. acquired funding.

## Competing interests

The authors declare no competing interests.
