## [Peer Review File · Nature Communications]

Reviewers' comments:

Reviewer #1 (Remarks to the Author):

In *Escherichia coli* two proteins – RelA and SpoT – control the intracellular pool of alarmone nucleotide (p)ppGpp. While our understanding of RelA has advanced dramatically over last years, SpoT remains unexplored. Therefore, further studies of SpoT are warranted and substantial advances in the topic are welcome.

In this report Germain and colleagues characterise a potential novel regulator of SpoT – a small (8 kD) protein YtfK, a member of the Pho regulon that is induced upon phosphate starvation. While SpoT is an essential gene and is a well-established key player in bacterial physiology, YtfK is a rather obscure protein. Currently, we do not have much phenotypic data. The available fitness data for the *ytfK* deletion strain suggest that the protein is important upon aluminum chloride stress, whatever that would mean, see <http://fit.genomics.lbl.gov/cgi-bin/singleFit.cgi?orgId=Keio&locusId=18242&showAll=0>

The authors convincingly show that YtfK is connected to induction of ppGpp accumulation in *E. coli*. They propose renaming the protein to TspT, standing for 'Trigger of SpoT'. Since I am not very convinced by the mechanistic aspects of the study, I will refer to the protein as YtfK. There are many technical issues with nearly all of the aspects of this work. Therefore, I suggest that the paper should be rejected with a possibility of a resubmission of a re-worked manuscript.

Specific comments:

1. In order to understand how general the potential regulation of SpoT by YtfK might be, the authors should map its distribution (gene presence/absence) in bacteria and compare it to that of SpoT. If I understand correctly, YtfK has very narrow taxonomic distribution.

2. YtfK was discovered as a protein that overcomes the inability of relaxed *E. coli* (*relA* knock-out) to grow on so-called SMG plates (Figure 1). The initial discovery was made using the ASKA collection of plasmids over-expressing 6His-tagged proteins. While this approach is well-suited for protein purification and initial genetic screens, there is no good reason to use a 6His-tagged protein for genetic testing that follows the initial hit identification, especially in the case of YtfK. It is a very small protein (8 kDa) and is expected to form complexes and regulate other proteins, therefore adding a tag is potentially problematic – and we see clear evidence of that in the data (see below). Therefore, the SMG phenotype tests presented on Figure 1 should be done with native, untagged protein. An essential (and currently missing) control is to show that YtfK expressed in trans rescues the *ytfK* knock-out phenotype, i.e. the genetic system works as expected.

3. There are clear signs of the genetic system being unstable and 6His-tagging affecting the results:

a) on Figure 4C the YtfK-6His is clearly toxic, more so in wt compared to the *relA* knock-out background; in wt, YtfK-6His completely abrogates the growth

b) but on Figures S1 and S3 untagged YtfK displays no toxicity on nutrient agar, suggesting that we are dealing with a gain-of-function toxicity upon addition of 6His

c) and then, surprisingly, on Figure S3 while both untagged and tagged proteins save *relA* knock-out on SMG, the 6His-tagged protein has no toxicity. This suggests that we are picking up compensatory mutations.

Therefore, a) it is absolutely essential that all the key experiments are done with untagged proteins and proper complementation controls are done (see 2., above) and b) the authors should report growth curves with tagged and untagged side by side.

4. Minor, but interesting observation: it is well-established, that spoT cannot be knocked out in *E. coli* when relA is present in the genome, unless compensatory mutations are picked up: in the absence of the (p)ppGpp hydrolytic activity of SpoT, RelA is highly toxic. Therefore, one would expect that in ppGpp⁰ (relA and spoT double knock-out) background IPTG-inducible expression of RelA should be very toxic. However, this effect is very weak, if at all present (Figure S1). The authors should comment on this.

5. TLC-based measurements of ppGpp levels upon starvation using ytfK, relA and ytfK + relA double knock-out strains (Figure 2). The authors use three starvation conditions:

Figure 2A, phosphate,

Figure 2B, fatty acid starvation induced by the antibiotic cerulenin and

Figure 2C: amino acid starvation induced by L-valine that causes amino acid biosynthesis imbalance leading to amino acid starvation.

Technical comment: the analyses were performed by TLC, experiments repeated at least three times and only one repetition is shown. The authors should quantify the spot intensity on original TLCs, generate quantitative data, calculate means and errors and plot the data as a scatter plot of ppGpp fraction vs time. This will generate 3 panels with 4 kinetic traces each, allowing analysis of the data. The current version does not allow assessment of the size of the effects nor reproducibility.

Phosphate and fatty acid starvations were reported both to cause SpoT activation, and amino acid starvation causes activation of RelA. The results presented in the current version of the manuscript are striking – but counterintuitive:

a) Importantly, since YtfK is a part of Pho regulon, phosphate limitation is the biologically relevant stress to test. The limitation induces ppGpp accumulation in ytfK knock out, suggesting that YtfK is not necessary for SpoT activation and cannot be the key regulator in this case. relA deletion has no effect, as expected. The combined addition of relA and ytfK almost abrogated ppGpp accumulation, suggesting that either RelA and YtfK together activate SpoT (a very exotic possibility), or YtfK activates RelA (also exotic, but readily testable biochemically) or that we have some indirect effects at play, e.g. the authors suggest in the discussion that ppGpp generated by SpoT activates RelA to amplify the signal (i.e. (p)ppGpp production). While pppGpp does, indeed, activate RelA in vitro, to become meaningfully strong, this the effect requires the presence of deacylated tRNA associated with ribosomal complexes (Kudrin et al. 2018), and therefore is unlikely to operate upon phosphate limitation.

Note that the relA-dependence of YtfK is reproducible, see Figure 4B: ppGpp accumulation upon expression of 6His-tagged YtfK is much stronger in the presence of RelA, compared with wild type vs relA knock-out backgrounds. So far the data really point towards YtfK activating RelA, an exciting possibility that should be tested by the authors biochemically.

b) In the case of cerulenin challenge (fatty acid starvation), the ytfK knock-out affected ppGpp accumulation relA-independently. This suggests that either YtfK can directly regulate SpoT either alone or together with RelA, or that we again have some indirect effects. Direct biochemical tests are essential to gain some clarity.

6. Two-hybrid system tests of YtfK interactions with RelA and SpoT. Importantly, the BTH101 strain used by authors encodes both native RelA and SpoT chromosomally. While endogenous RelA is a relatively abundant protein and is, therefore, likely to compete well with pKT25relA, SpoT is exceedingly scarce and, therefore, will not be able to compete with pKT25spoT. This likely invalidates the specificity of the assay, and, therefore, ppGpp⁰ BTH101 strain should be used.

A good control used by the authors is the P42L YtfK mutant that does not interact with SpoT

according to two-hybrid and is not toxic. Unfortunately, the authors do not use this mutant to its full potential, and the proposed SpoT:YtfK interaction should be tested biochemically using P42L YtfK mutant alongside with the wild type YtfK (Figure 5B, see specific comments below, 8.).

7. Technical note on in silico modelling (Figure 5c): modelling a complex of two predicted structures with no constraints is a really long shot and can be misleading. If anything I would keep this figure as a supplementary (or, preferably, delete).

8. Binding studies and (currently lacking) biochemical assays. RelA SpoT Homologue proteins are challenging targets and tagging has been shown to interfere with their function – even adding a 6His, see Kudrin et al. NAR 2018. Therefore it is absolutely essential that the proteins used for biochemical assays are well-characterised. SpoT is a very challenging protein for biochemistry and, as long RSHs tend to, tends to precipitate. Truncated versions are more soluble, but the use of truncates limits experiments to meaningless biochemical investigations. Therefore the key results generated by extensive experiments with truncates should be validated with full-length proteins.

a) In order to interpret the gel-filtration profiles (Figure S5) one needs to add a calibration experiment with Mw standards. The authors use truncated SpoT with a Mw of around 40 kDa (the SDS PAGE is provided). At the same time, on the HiLoad 16/600 Superdex 200 column the protein elutes at around 60 mL, between Ferritin (Mw 440 kDa) and Aldolase (Mw 158 kDa), while one would expect it to run as Ovalbumin (Mw 43 kDa), around 80 ml. This suggests that the protein aggregates compromising the follow-up experiments. Therefore it is essential that

- 1) calibration experiments for gel-filtration are provided
- 2) if the gel-filtration results are ambiguous, alternative approaches such as dynamic light scattering are used to assess the aggregation state.

b) The interaction between YtfK and truncated SpoT has very low affinity, a Kd of 16 μ M. Given that there are very, very few SpoT molecules in the cell, the Kd in this range is likely to be meaningless. Therefore,

- 1) experiments with full-length proteins are needed
- 2) P42L YtfK mutant should be used as a control.

c) The authors suggest that YtfK activates SpoT and they have purified the proteins. Therefore, they need to perform the enzymatic assays to directly show that the interaction is meaningful and, indeed, leads to activation of SpoT's synthetic activity

d) Since YtfK seems to activate RelA (Figure 4B and Figure 2B, and also see comments regarding specificity of the two-hybrid assays), this should be tested – biochemical assays with RelA are readily available. This experiment is essential since the authors base the whole story on YtfK being a novel Trigger of SpoT (and not RelA).

9. The biological role of YtfK in antibiotic tolerance (Figure 3). I find the title of the figure is misleading: 'TspT mediates antibiotic tolerance'. What we are looking at is increased sensitivity of knock-out strains. These mutant strains become more sensitive when they are compromised, but this does not imply that antibiotic tolerance is the function or role of YtfK: the fact that expression is induced upon phosphate limitation – and no reports of induction upon antibiotic challenge – suggests that the role as something to do with regulation of phosphate metabolism, and it is not a dedicated antibiotic tolerance mechanism. I find it strange that the kill curves are performed using bacteria challenged with fatty acid limitation, but not upon phosphate starvation. The latter is more likely to be relevant, since, as mentioned earlier, YtfK is a member of the Pho regulon. While authors do embrace the role of YtfK in phosphate metabolism – Line 231 'We show that TspT is essential to maintain normal (p)ppGpp level in response to both fatty acid and phosphate starvation (Figure 2). This regulation seems particularly important for cell survival' – no experiments were performed to directly test this.

a) General technical comment: corresponding growth curves are essential to make sense of the ampicillin kill curves, since without growth curves it is impossible to rationalise the effects of ampicillin challenge: the efficiency of AMP is a function of growth.

b) We need phosphate starvation experiments directly addressing the importance of YtfK upon survival during phosphate starvation. Experiments could be performed both in the absence and in presence of a beta-lactam challenge.

c) The authors suggest that YtfK is essential for SpoT (p)ppGpp synthetic activation and state that the lack of viability after cerulenin treatment is tied to (p)ppGpp. If this is the case, why are there are 2 logs higher % survival after cerulenin treatment in the *relA ytfK* double knock-out (where SpoT should not be activated) compared to the *relA spoT* double knock-out strain? This clearly suggests that SpoT is able to be activated independently of TspT, and that therefore the role of YtfK as an activator is clearly over-stated.

d) On Figure 3A the *ytfK* knock-out strain shows no defect in viability after cerulenin treatment whereas the *relA ytfK* double knock-out does. Conflictingly with the hypothesis of ppGpp-mediated tolerance governed by YtfK, on Figure 2B, the *ytfK* knock-out and *relA ytfK* double knock-out strains produce seemingly similar amounts of (p)ppGpp upon cerulanin treatment. This clearly suggests that the level of (p)ppGpp is not the deciding factor in viability.

e) Note that the relationships between cerulenin, (p)ppGpp and ampicillin tolerance have been extensively investigated by the lab of Edward Ishiguro, who concluded using ppGpp0 strains that 'Penicillin tolerance was shown to be a direct consequence of the inhibition of phospholipid synthesis and not due to the possible accumulation of guanosine-3',5'-bispyrophosphate (ppGpp), the starvation stress signal molecule known to be responsible for the development of penicillin tolerance in amino-acid- deprived bacteria.' (Rodionov and Ishiguro 1996). No references to Ishiguro's works in the paper by Germain and colleagues. I suggest amending this, and since there are clear differences in ppGpp0 behaviour upon cerulenin challenge in Ishiguro's lab and in the current paper, it is prudent to check if the strain used in the current work is not contaminated by phages (see works by Maisonneuve and colleagues recently retracted from PNAS and Cell)

f) Line 160 'These results strongly suggest that, indeed, elevated (p)ppGpp level confers antibiotic tolerance' I feel that the data are not strong enough to provide the evidence for this statement. The inhibition of FAS by cerulenin treatment renders cells non-growing due to depletion of phospholipid precursors, this should not be dependent on (p)ppGpp production. This in its self should confer tolerance to ampicillin? (p)ppGpp is produced in response to the stress and aids adaptation of the cell and cell survival, i. e. can you say from this data that this is the driving force that confers antibiotic tolerance? Again, growth curves are essential.

10. General comment on data presentation: The authors use various constructs in various many experiments. It is essential that it is clear which strains and plasmids are used where, this needs to be unambiguously specified in figure legends with a reference to plasmid and strain table. Similarly, different concentrations of IPTG are used in various figures: 1 mM on Figure 1, 500 uM on Figure 4 and 200 uM on Figure 3S. This unnecessarily complicates direct comparisons.

To conclude: starting from the Abstract, the authors rename YtfK to TspT, Trigger of SpoT (and the abbreviation TspT is first spelled out on line 110). At this moment the results are not solid enough: Trigger of SpoT AND RelA is a possible interpretation as well. I feel that trademarking the data is premature at this point.

Reviewer #2 (Remarks to the Author):

In this very important work, the authors report the discovery of a new factor, TspT, required for the ppGpp synthase activity of SpoT in Escherichia coli. The findings are compelling, since in a tspT mutant, there is no ppGpp apparition in stress conditions known to normally trigger SpoT-dependent ppGpp accumulation. And in reverse, the overproduction of TspT is enough to trigger ppGpp increase. This finding is of primary importance in the field, given the role of ppGpp in survival and antibiotics resistance, and given that we know so little about the mechanism of control of the enzymes of the SpoT family.

General comments

The experiments clearly demonstrate the requirement of TspT for SpoT synthase activity. However, it is not clear if the role of TspT is really to detect starvation and play a role in the regulation of SpoT per se, or if simply TspT presence is needed for SpoT synthase activity. The authors show the effect of TspT in conditions of phosphate starvation and fatty acid starvation, but it is very easy to imagine that the same effect might be observed for any conditions triggering ppGpp synthesis in the absence of RelA.

If there is really a regulation of SpoT activity by TspT, then it is likely to be acting by a change in TspT levels (suggested by the activation obtained simply by TspT overproduction). There are published data about the regulation of tspT expression by the PhoB transcriptional activator. Does the observations reported here correlated with changes in the amounts of TspT protein? What are the ratio of TspT and SpoT proteins in the cell in balanced conditions or during stress response ? Such information is important to interpret the kinetic constant measured for the TspT/SpoT interaction in vitro. It would also be interesting to get an idea of the levels of expression (tspT gene expression or TspT protein levels) in the different conditions tested. Also, the results of the 2017 paper on TspT should be discussed in regard of the results of the present work.

It would be important to give rigorous information about the conservation of tspT in bacteria. Is TspT systematically present in bacteria having two distinct RSH enzymes SpoT and RelA ? Is it present in other types of bacteria ?

Specific comments

In the discussion, the authors show a figure of a 3D modelling of the potential TspT-SpoT complex, based on the structure of RelSeq enzyme and a structure of TspT available in the pdb. However, there is very little information on how the model was obtained (parameters, assessment of the quality of the final model for example), and is it not necessary to have access to the structure file? Furthermore, there is very little use of this model in the paper, for example the position of the Proline 42 of TspT shown to be important for the interaction with SpoT is not even shown on the figure or discussed.

Concerning the 2hybrid : the results are overall very clear and convincing. However, it is important to show the correct expression of the diverse hybrid proteins (especially for constructs that do not interact). In addition, the results about specificity of interaction (no interaction with RelA) are shown in only one combination of vectors. Same for the identification of the domain of interaction. The authors should at least explain why they choose a combination rather than the other.

Concerning the in vitro binding assay : because the Kd obtained is rather low, the YtfK(P42L) mutant, and importantly the catalytic domain of RelA should be used as controls.

Minor comments

The strain constructions might be described a bit more (how the kanaR cassette was removed ? describe the two successive P1 transduction for ppGpp^o strain construction for example etc...).

There is a very lengthy description of the methods for protein production and purification, while other equally important parts are overly succinctly described or missing (initial screening for tspT and mutagenesis screen on tspT; 3D modelling; strain construction). For example, for the screen using the ASKA collection, the description of the expression clones is important : are the genes tagged, with 6his or with GFP ?

figure S2 : what is the meaning of -ve/+ve ? If it means "empty vector", why is it sometimes negative, sometimes positive? What is the difference between the left panels and the right panels ? They seem completely redundant.

Reviewer #3 (Remarks to the Author):

The stringent response is a conserved regulatory mechanism allowing bacteria to adapt to a variety of stressful conditions. Recently, major new insights were gained into the molecular mechanisms. Here the author made a further major contribution showing that the enzymatic activity of the bifunctional enzyme SpoT is modulated through interaction with a small cytoplasmic protein, here named TspT. This is an interesting finding which shed new lights onto the long-lasting question how SpoT contributes to stringent response in *E. coli*. The manuscript is well written. However at some points a more thorough analysis could help to get a more complete picture. Some questions can be addressed with the tools in hands. e.g. How is Tsp regulated under fatty acid starving conditions? Does Tsp interaction with SpoT contribute to H₂O₂ tolerance? How does TspT influence the synthetase/hydrolase activity of SpoT in vitro?

Specific comments

Line 28 and throughout: Since the identified SpoT interacting protein already has a designation (YtfK) and was already shown to be regulated by Pho as well as to be involved in oxidative stress survival I suggest to stick to YtfK instead of renaming it to TspT.

Figure 2C: As it seems YtfK/SpoT induces only ppGpp not pppGpp. This should be mentioned and discussed.

Line 117, Figure 2A: Because the link of phosphate starvation and ppGpp (via TSP) is central to the whole story, quantitate results based on the three independent biological replicates should be shown.

Line 155, Figure 2 B: The GTP spot as indicated is confusing. Why GTP is only visible after Cerulenin addition?

Line 145 and following, Figure 3: It should be indicated whether there are any differences in growth or MIC (cerulelin, ampicillin) between the strains analysed? For results shown in figure 3 B and 3 C the relA/spoT and relA/tsp mutant should be included in the analyses, to show that the effects are due to TSP mediated ppGpp synthesis. This mutant should have the same or at least very similar phenotype to the tsp mutant. It would be also helpful to know whether TspT expression is increased by cerulelin. YtfK/Tsp was already shown to be involved in tolerance towards H₂O₂ and phosphate starvation (Iwadate Y, Kato JI. Microbiology. 2017;163:1912). It would be very interesting to see whether this also mediated via SpoT.

Line 10: Since the authors already have purified Tsp and SpoT in hand it should be feasible to perform an in vitro activity assay to finally confirm that Tsp activates ppGpp synthesis or possibly inhibits ppGpp degradation.

Discussion:

Some more information on YtfK/TspT should be given. Is it conserved in different bacteria? Localisation, basic protein? It was recently shown that YtfK is involved in H₂O₂ tolerance (Iwadate Y and Kato J, Microbiology). This should be at least discussed. Is there anything known about the

regulation of yftK besides being part of the Pho regulon.

Line 340: What is meant with suppress or complement the growth defect? You probably screened for growth

Reviewers' comments:

Reviewer #1 (Remarks to the Author):

In *Escherichia coli* two proteins – RelA and SpoT – control the intracellular pool of alarmone nucleotide (p)ppGpp. While our understanding of RelA has advanced dramatically over last years, SpoT remains unexplored. Therefore, further studies of SpoT are warranted and substantial advances in the topic are welcome.

In this report Germain and colleagues characterise a potential novel regulator of SpoT – a small (8 kD) protein YtfK, a member of the Pho regulon that is induced upon phosphate starvation. While SpoT is an essential gene and is a well-established key player in bacterial physiology, YtfK is a rather obscure protein. Currently, we do not have much phenotypic data. The available fitness data for the ytfK deletion strain suggest that the protein is important upon aluminum chloride stress, whatever that would mean, see

<http://fit.genomics.lbl.gov/cgi-bin/singleFit.cgi?orgId=Keio&locusId=18242&showAll=0>

The authors convincingly show that YtfK is connected to induction of ppGpp accumulation in *E. coli*. They propose renaming the protein to TspT, standing for ‘Trigger of SpoT’. Since I am not very convinced by the mechanistic aspects of the study, I will refer to the protein as YtfK. There are many technical issues with nearly all of the aspects of this work. Therefore, I suggest that the paper should be rejected with a possibility of a resubmission of a re-worked manuscript.

AU: The severe criticism from Reviewer 1 confuses us. It has taken several years to develop reliable, stable and robust genetics tools to address the difficult of how *E. coli* controls its intracellular level of the second messenger (p)ppGpp.

We hope that the additional experiments and improvements of the text will be sufficient to convince Reviewer 1 of the robustness of the genetic tools and the general interest of our work.

Specific comments:

1. In order to understand how general the potential regulation of SpoT by YtfK might be, the authors should map its distribution (gene presence/absence) in bacteria and compare it to that of SpoT. If I understand correctly, YtfK has very narrow taxonomic distribution.

AU: We thank the referee for this important remark. While the questions related to the distribution and evolution of YtfK is of important interest, we believe that the fact that the YtfK coding sequence is rather short renders difficult the identification of small proteins orthologs. To date YtfK seems specific to Gammaproteobacteria with distribution in the orders of *Enterobacteriales*, *Vibrionales*, *Aeromonadales*, *Pasteurellales* and *Alteromonadales*¹. This information is now provided in the discussion of the revised manuscript.

2. YtfK was discovered as a protein that overcomes the inability of relaxed *E. coli* (*relA* knock-out) to grow on so-called SMG plates (Figure 1). The initial discovery was made using the ASKA collection of plasmids over-expressing 6His-tagged proteins. While this approach is well-suited for protein purification and initial genetic screens, there is no good reason to use a 6His-tagged protein for genetic testing that follows the initial hit identification, especially in the case of YtfK. It is a very small protein (8 kDa) and is expected to form complexes and regulate other proteins, therefore adding a tag is potentially problematic – and we see clear evidence of that in the data (see below). Therefore, the SMG phenotype tests presented on Figure 1 should be done with native, untagged protein. An essential (and currently missing) control is to show that YtfK expressed in trans rescues the *ytfK* knock-out phenotype, i.e. the genetic system works as expected.

AU: This comment confuse us because the original data presented in Figure 1 and S1A were done with native and untagged protein as mentioned in the Figure 1 and S1A and also in Figure legends. We nevertheless added a sentence in the result section of the revised manuscript mentioning that *ytfK* was cloned in the pEG25 plasmid under an IPTG inducible promoter. We performed additional experiments showing that *ytfK* in Trans suppresses the growth defect of a *ΔrelA ytfK* strain on SMG medium. The results are shown in Figure S1B of the revised manuscript.

3. There are clear signs of the genetic system being unstable and 6His-tagging affecting the results:

a) on Figure 4C the YtfK-6His is clearly toxic, more so in wt compared to the *relA* knock-out background; in wt, YtfK-6His completely abrogates the growth

b) but on Figures S1 and S3 untagged YtfK displays no toxicity on nutrient agar, suggesting that we are dealing with a gain-of-function toxicity upon addition of 6His

c) and then, surprisingly, on Figure S3 while both untagged and tagged proteins save *relA* knock-out on SMG, the 6His-tagged protein has no toxicity. This suggests that we are picking up compensatory mutations.

Therefore, a) it is absolutely essential that all the key experiments are done with untagged proteins and proper complementation controls are done (see 2., above) and b) the authors should report growth curves with tagged and untagged side by side.

AU: The referee is correct. Adding a tag can be potentially problematic specially by stabilizing the protein. Therefore for the sake of clarity we performed additional experiments with untagged version and **we have replaced all previous figures** presenting results obtained *in vivo* with his-tagged YtfK (see new Figure 1C; 5B-C-D of the revised manuscript).

Moreover, we provide complementation experiments with *ytfK* cloned in the pEG220 plasmid under native promoter (Figure 4 of the revised manuscript) and finally we also provide growth curves in the revised manuscript (Figure 4 and Figure 5).

4. Minor, but interesting observation: it is well-established, that *spoT* cannot be knocked out in *E. coli* when *relA* is present in the genome, unless compensatory mutations are picked up: in the absence of the (p)ppGpp hydrolytic activity of SpoT, RelA is highly toxic. Therefore, one would expect that in ppGpp⁰ (*relA* and *spoT* double knock-out) background IPTG-inducible expression of RelA should be very toxic. However, this effect is very weak, if at all present (Figure S1). The authors should comment on this.

AU: We agree with the referee that such observations can be misleading at first glance. The construction details of the plasmid harboring *relA* was not sufficiently described in the manuscript and we apologized for this omission. We have improved the supplementary material and methods section accordingly.

Indeed it is correct that SpoT cannot be deleted in *E. coli* when wt copy of *relA* is presented at the proper locus. To circumvent this problem and therefore be able to express *relA* in the $\Delta relA$ *spoT* strain we have engineered a robust system to tightly regulate intracellular level of RelA.

- First we have constructed the pEG25 plasmid which contains the Pt5 promoter from the pCA24N. This promoter contains three Lac operators and is therefore less leaky. As shown from Figure S1 of the revised manuscript, when *relA* is cloned on the pEG25 plasmid the leakiness of the promoter is not sufficient to complement the $\Delta relA$ growth defect on SMG medium.

-Most importantly we played with constants that contribute with the process of translation initiation. Indeed by engineering the Shine-Dalgarno sequence, the initiation codon and the spacer between them, we were able to tightly regulate intracellular level of RelA. The chosen Shine-Dalgarno sequence is AGGA, the initiation codon is UUG and the distance between them is 4 bases. The translation initiation of this construct is predicted to be two to three orders of magnitude lower than the wt chromosomal encoded *relA* locus².

Using this construct we observed that RelA complements the $\Delta relA$ growth defect on SMG medium in an IPTG dependent manner and become toxic on nutrient agar, SMG and minimal medium plates when overexpressed in the $\Delta relA$ *spoT* strain at concentration above 100 μ M (Figure S1 of the revised manuscript) . Therefore the system is extremely robust and genetically stable.

5. TLC-based measurements of ppGpp levels upon starvation using *ytfK*, *relA* and *ytfK* + *relA* double knock-out strains (Figure 2). The authors use three starvation conditions:

Figure 2A, phosphate,

Figure 2B, fatty acid starvation induced by the antibiotic cerulenin and

Figure 2C: amino acid starvation induced by L-valine that causes amino acid biosynthesis imbalance leading to amino acid starvation.

Technical comment: the analyses were performed by TLC, experiments repeated at least three times and only one repetition is shown. The authors should quantify the spot intensity on original TLCs, generate quantitative data, calculate means and errors and plot the data as a scatter plot of ppGpp fraction vs time. This will generate 3 panels with 4 kinetic traces each, allowing analysis of the data. The current version does not allow assessment of the size of the effects nor reproducibility.

AU: We apologize for the omission of compilation of aggregate data in our manuscript at the first stage of submission. We now provide quantification of (p)ppGpp accumulation in the three starvation conditions but also for experiments with overproduction of YtfK and YtfK^{P42L} (Figure 3 and 5 respectively in the revised manuscript).

Phosphate and fatty acid starvations were reported both to cause SpoT activation, and amino acid starvation causes activation of RelA. The results presented in the current version of the manuscript are striking – but counterintuitive:

a) Importantly, since YtfK is a part of Pho regulon, phosphate limitation is the biologically relevant stress to test. The limitation induces ppGpp accumulation in *ytfK* knock out, suggesting that YtfK is not necessary for SpoT activation and cannot be the key regulator in this case.

AU: We do not agree with the referee remark regarding the absence of biological role of YtfK in the accumulation of ppGpp observed during phosphate starvation. Indeed as shown in Figure 3 of the revised manuscript we observed a significant reduction of ppGpp in the $\Delta ytfK$ mutant. In addition our new experiments provided in Figure S3 show that YtfK is also produced during fatty acid starvation arguing that the biological relevance of YtfK is not limited to phosphate starvation.

relA deletion has no effect, as expected. The combined addition of *relA* and *ytfK* almost abrogated ppGpp accumulation, suggesting that either RelA and YtfK together activate SpoT (a very exotic possibility), or YtfK activates RelA (also exotic, but readily testable biochemically) or that we have some indirect effects at play, e.g. the authors suggest in the discussion that ppGpp generated by SpoT activates RelA to amplify the signal (i.e. (p)ppGpp production). While pppGpp does, indeed, activate RelA in vitro, to become meaningfully strong, this the effect requires the presence of deacylated tRNA associated with ribosomal complexes (Kudrin et al. 2018), and therefore is unlikely to operate upon phosphate limitation.

AU: We agree with the referee and apologized for the non-correct statement of this sentence. This sentence has been removed and discussion has been changed accordingly.

Note that the *relA*-dependence of YtfK is reproducible, see Figure 4B: ppGpp accumulation upon expression of 6His-tagged YtfK is much stronger in the presence of RelA, compared with wild type vs *relA* knock-out backgrounds. So far the data really point towards YtfK activating RelA, an exciting possibility that should be tested by the authors biochemically.

AU: We indeed confirm that RelA contributes to the accumulation of ppGpp during phosphate starvation and fatty acid starvation (see quantification from the revised Figure 3A and 3B) but also when we overexpressed *ytfK* (Figure 5B-C) and we believe that it is rather indirect by residual activation of RelA as discussed in the revised manuscript.

Indeed it is well established that the hydrolysis function of SpoT is essential for balancing cellular (p)ppGpp concentrations in the presence of RelA, and disruption of the *spoT* gene in *E. coli* is therefore lethal³ even under amino acid replete conditions. Moreover it is currently admitted that RSHs avoid to simultaneously synthesize and degrade (p)ppGpp primarily to prevent futile cycle⁴. Thus, we suggest that YtfK specifically pushes the catalytic balance of SpoT toward (p)ppGpp synthesis rather than hydrolysis therefore enabling residual activation of RelA to maximize the alarmone production.

Note that the RelA-dependent contribution of ppGpp accumulation when YtfK is overproduced is not the result of uncharged tRNA accumulation arising from rapid consumption of amino acid pools due to YtfK overproduction. Indeed as observed, the YtfK^{P42L} variant which is produced at a similar level, does not induce accumulation of ppGpp in wt strain (Figure 5B-C and S6B-C).

Finally, despite strong arguments supporting the specific role of YtfK on SpoT activities, we indeed cannot totally rule out that YtfK does not activate RelA directly. Therefore we decided not to rename YtfK and we have changed the discussion accordingly.

b) In the case of cerulenin challenge (fatty acid starvation), the *ytfK* knock-out affected ppGpp accumulation *relA*-independently. This suggests that either YtfK can directly regulate SpoT either alone or together with RelA, or that we again have some indirect effects. Direct biochemical tests are essential to gain some clarity.

AU: The provided quantification in the revised manuscript (Figure 3B) show that there is a RelA-dependent and a RelA-independent accumulation (consistent to what explained above).

Note that recent report suggests that fatty acid starvation leads to depletion of lysine that, in turn, leads to the accumulation of uncharged tRNA^{Lys} and activation of RelA⁵. However exogenous addition of lysine suppresses activation of RelA during fatty acid starvation⁵. Given that our experimental setup include all 20 amino acid we do not believe that such activation occurs in our conditions.

6. Two-hybrid system tests of YtfK interactions with RelA and SpoT. Importantly, the BTH101 strain used by authors encodes both native RelA and SpoT chromosomally. While endogenous RelA is a relatively abundant protein and is, therefore, likely to compete well with pKT25relA, SpoT is exceedingly scarce and, therefore, will not be able to compete with pKT25spoT. This likely invalidates the specificity of the assay, and, therefore, ppGpp0 BTH101 strain should be used.

AU: We acknowledge the referee for this interesting remark. The lactose operon (and also the maltose operon) is subjected to regulation by other signals in addition to cAMP. Unfortunately, (p)ppGpp is required for maximal transcription of the lac operon^{6,7}. Therefore a ppGpp0 Δ cyaA is not suitable for two hybrid approach.

Nevertheless we performed additional experiments to address the referee concern.

As shown in the Figure S5 of the revised manuscript we failed to detect an interaction between RelA and YtfK even when *relA* is cloned in the pUT18C (or pUT18) which is a high copy number plasmid. In addition we do not detect any interaction between YtfK and RelA in the all 8 combinations and orientations tested. Finally using BTH assay we observed that RelA multimerizes *in vivo* in the 4 tested combinations as previously observed⁸.

Therefore our results show that the RelA fusions are correctly expressed and that in such assay the endogenous RelA is not able to compete with RelA-fusions encoded from pKT25 or pUT18 derivatives plasmids.

A good control used by the authors is the P42L YtfK mutant that does not interact with SpoT according to two-hybrid and is not toxic. Unfortunately, the authors do not use this mutant to its full potential, and the proposed SpoT:YtfK interaction should be tested biochemically using P42L YtfK mutant alongside with the wild type YtfK (Figure 5B, see specific comments below, 8.).

AU: We agree with the referee. Unfortunately we spent a lot of time trying to purify it without success. The YtfK P42L protein appears unstable *in vitro* and rapidly aggregate during the thioredoxin-tagged cleavage.

7. Technical note on in silico modelling (Figure 5c): modelling a complex of two predicted structures with no constraints is a really long shot and can be misleading. If anything I would keep this figure as a supplementary (or, preferably, delete).

AU: The referee is correct. To define Haddock run restraints, we used the surface exposed residues of both YtfK and SpoT as active residues that could be directly involved in the interaction. For YtfK, we selected the surface exposed residues manually using PyMol. Based on the specificity of YtfK-SpoT interaction we selected surface exposed residues in SpoT that are not conserved in RelA. As for YtfK, the SpoT residues were manually selected using PyMol. We apologize for this omission and have corrected the materials and methods accordingly.

Finally we followed the referee recommendation and moved the FigureS8 in supplementary information of the revised manuscript.

8. Binding studies and (currently lacking) biochemical assays. RelA SpoT Homologue proteins are challenging targets and tagging has been shown to interfere with their function – even adding a 6His, see Kudrin et al. NAR 2018. Therefore it is absolutely essential that the proteins used for biochemical assays are well-characterised. SpoT is a very challenging protein for biochemistry and, as long RSHs tend to, tends to precipitate. Truncated versions are more soluble, but the use of truncates limits experiments to meaningless biochemical investigations. Therefore the key results generated by extensive experiments with truncates should be validated with full-length proteins.

AU: We agree with the referee that truncated version limits *in vitro* experiments, in particular reconstitution of biochemical activity. We also agree that full length SpoT is a very challenging protein for biochemistry assay. Therefore and as explained below (point 8c) we do not believe that enzymatic assay is needed for the story to be complete. Moreover given that the strength of the interaction between YtfK-SpoT¹⁻³⁷⁸ and the YtfK-SpoT is similar *in vivo*, we do not believe that *in vitro* interaction between YtfK and SpoT (full length) is required for this story.

a) In order to interpret the gel-filtration profiles (Figure S5) one needs to add a calibration experiment with Mw standards. The authors use truncated SpoT with a Mw of around 40 kDa (the SDS PAGE is provided). At the same time, on the HiLoad 16/600 Superdex 200 column the protein elutes at around 60 mL, between Ferritin (Mw 440 kDa) and Aldolase (Mw 158 kDa), while one would expect it to run as Ovalbumin (Mw 43 kDa), around 80 ml. This suggests that the protein aggregates compromising the follow-up experiments. Therefore it is essential that

- 1) calibration experiments for gel-filtration are provided
- 2) if the gel-filtration results are ambiguous, alternative approaches such as dynamic light scattering are used to assess the aggregation state.

AU: The referee is correct and we are grateful for this suggestion. Calibration experiment for gel filtrations have been performed and are now provided in Figure S7.

As shown in the previous version of the manuscript SpoT¹⁻³⁷⁸ indeed elutes around 60 mL, between Ferritin (Mw 440 kDa) and Aldolase (Mw 158 kDa) but at a very different volume from the void volume of the column where we expected to find aggregated material.

Importantly in our previous purification protocol we used Triton X-100 and believe that the aberrant Mw observed in our gel filtration was the result of Triton micelles interaction with YtfK.

To circumvent this problem we have modified our purification protocol by removing Triton from all purification steps and by adding CHAPS only in the lysis buffer (see material and methods of the revised manuscript). Doing that we obtained soluble SpoT¹⁻³⁷⁸ with an apparent Mw on gel filtration around 80000 kDa (suggesting dimeric form). Dynamic light scattering analysis confirm this apparent molecular weight.

We therefore performed additional blitz experiments with newly purified SpoT¹⁻³⁷⁸ using this protocol. We are now presenting triplicate titration experiment (with standard deviation) for 4 different concentrations of SpoT¹⁻³⁷⁸ (see Figure 7A of the revised manuscript).

Finally we would like to thank the referee for requesting DLS experiments. Indeed, using this alternative approaches we not only confirm the protein quality (absence of aggregation) but we were also able to further confirm and analyze the YtfK- SpoT¹⁻³⁷⁸ interaction.

Remarkably, when YtfK is titrated into SpoT¹⁻³⁷⁸, the size of SpoT¹⁻³⁷⁸ is reduced. These results presented in Figure 7B of the revised manuscript further support that YtfK triggers conformational changes in the catalytic domain of SpoT.

b) The interaction between YtfK and truncated SpoT has very low affinity, a K_d of 16 μM. Given that there are very, very few SpoT molecules in the cell, the K_d in this range is likely to be meaningless. Therefore,

- 1) experiments with full-length proteins are needed
- 2) P42L YtfK mutant should be used as a control.

AU: We agree that the apparent K_D obtained *in vitro* by using Biolayer interferometry suggested indeed a weak and transient/ highly dynamic interaction (Figure 7A of the revised manuscript). It is worth pointing that weak interactions are also crucial for cellular organization and functions. The referee points to the few numbers of SpoT molecules in the cells to lower the importance of this interaction *in vivo*. Many different parameters, present in *E. coli* cytoplasm may affect the K_D *in vivo* (molecular crowding, additional partners...). For instance the cellular localization of SpoT, which is contradictory in the literature (membrane, ribosome, and cytosol) may affect the local concentration.

We therefore believe that the question is rather to understand if this interaction is physiologically productive for the cell (*ie* specific). Our *in vivo* experiments presented in the manuscript convincingly show that it is the case.

Unfortunately and as mentioned above the YtfK^{P42L} protein is unstable *in vitro* and rapidly aggregate during thioredoxin-tagged cleavage and cannot be used *in vitro*. Moreover the strength of the interaction between YtfK-SpoT¹⁻³⁷⁸ and the YtfK-SpoT appears similar *in vivo*, therefore we do not believe that *in vitro* interaction between YtfK and SpoT (full length) is required for this story.

c) The authors suggest that YtfK activates SpoT and they have purified the proteins. Therefore, they need to perform the enzymatic assays to directly show that the interaction is meaningful and, indeed, leads to activation of SpoT's synthetic activity

AU: It is our sincere opinion that such results are not required for our story to be complete and might be more appropriate in a fully independent story on the molecular/structural mechanisms behind the regulation of SpoT by YtfK. Moreover it might be realized that such enzymatic *in vitro* reconstitution sounds much harder than it looks.

d) Since YtfK seems to activate RelA (Figure 4B and Figure 2B, and also see comments regarding specificity of the two-hybrid assays), this should be tested – biochemical assays with RelA are readily available. This experiment is essential since the authors base the whole story on YtfK being a novel Trigger of SpoT (and not RelA).

AU: Even if we cannot rule out a direct activation of RelA by YtfK we believe that the contribution of RelA is rather indirect (as mentioned above). This opinion is based from three different observations:

-YtfK does not interact with full length RelA nor with the catalytic domain of RelA *in vivo* by two hybrid assays (as explained above; see FigureS5B-C-D).

- The (p)ppGpp level after amino acid starvation is not affected in a *ytfK* knock out strain (Figure 3C of the revised manuscript).

- The level of YtfK (which seems important for triggering the stringent response) is not changed during amino acid starvation supporting the hypothesis that YtfK does not have physiological role during amino acid starvation (Figure S3C of the revised manuscript).

Therefore if YtfK activates RelA directly, its role during amino acid starvation is meaningless. However, despite strong arguments supporting the role of YtfK on SpoT activities, we cannot totally rule out that YtfK does not activate RelA directly, therefore we decided not to rename YtfK and we change the discussion accordingly.

Finally and as mentioned above enzymatic reconstitution assay is a tricky experiment and is a complex one. To our knowledge only one group succeed at purifying untagged and functional RelA *in vitro*⁹.

9. The biological role of YtfK in antibiotic tolerance (Figure 3). I find the title of the figure is misleading: 'TspT mediates antibiotic tolerance'. What we are looking at is increased sensitivity of knock-out strains. These mutant strains become more sensitive when they are compromised, but this does not imply that antibiotic tolerance is the function or role of YtfK: the fact that expression is induced upon phosphate limitation – and no reports of induction upon antibiotic challenge – suggests that the role as something to do with regulation of phosphate metabolism, and it is not a dedicated antibiotic tolerance mechanism. I find it strange that the kill curves are performed using bacteria challenged with fatty acid limitation, but not upon phosphate starvation. The latter is more likely to be relevant, since, as mentioned earlier, YtfK is a member of the Pho regulon. While authors do embrace the role of YtfK in phosphate metabolism – Line 231 'We show that TspT

is essential to maintain normal (p)ppGpp level in response to both fatty acid and phosphate starvation (Figure 2). This regulation seems particularly important for cell survival' – no experiments were performed to directly test this.

a) General technical comment: corresponding growth curves are essential to make sense of the ampicillin kill curves, since without growth curves it is impossible to rationalise the effects of ampicillin challenge: the efficiency of AMP is a function of growth.

AU: The referee is correct and we are grateful for this suggestion. We have provided growth curves in Figure 4B and S4 of the revised manuscript.

b) We need phosphate starvation experiments directly addressing the importance of YtfK upon survival during phosphate starvation. Experiments could be performed both in the absence and in presence of a beta-lactam challenge.

AU: YtfK was already shown to be involved in tolerance towards phosphate starvation¹⁰. Finally for technical reason we decided to not address the survival during phosphate starvation in presence of ampicillin. Indeed kinetic of starvation for phosphate often fluctuates from days to days (mainly due to washing steps) rendering difficult the strict standardization needed for such experiment.

c) The authors suggest that YtfK is essential for SpoT (p)ppGpp synthetic activation and state that the lack of viability after cerulenin treatment is tied to (p)ppGpp. If this is the case, why are there are 2 logs higher % survival after cerulenin treatment in the *relA ytfK* double knock-out (where SpoT should not be activated) compared to the *relA spoT* double knock-out strain? This clearly suggests that SpoT is able to be activated independently of TspT, and that therefore the role of YtfK as an activator is clearly over-stated.

AU: we thanks the referee for this comment.

We performed additional experiment by diluting down 1/1000 bacterial culture before the addition of cerulenin antibiotics. Therefore the cells experienced 7-8 divisions before the addition of antibiotics and are therefore balanced. Doing that we observed that the viability of the *ΔrelA ytfK* strains after cerulenin treatment was further affected. The figure 4A of the revised manuscript has been change accordingly. However we agree that we still observed a significant 25 fold difference between the percentages of survival after cerulenin treatment in the *ΔrelA ytfK* compared to the *ΔrelA spoT* double knock-out strain. Therefore we indeed agree that SpoT is able to be activated in absence of YtfK. Importantly, our results show that 99.99% of cerulenin tolerant cells in the WT strain are RelA and YtfK-dependent pointing a very low contribution of SpoT alone (0.01%).

d) On Figure 3A the *ytfK* knock-out strain shows no defect in viability after cerulenin treatment whereas the *relA ytfK* double knock-out does. Conflictingly with the hypothesis of ppGpp-mediated tolerance governed by YtfK, on Figure 2B, the *ytfK* knock-out and *relA ytfK* double knock-out strains produce seemingly similar amounts of (p)ppGpp upon cerulanin treatment. This clearly suggests that the level of (p)ppGpp is not the deciding factor in viability.

AU. We do not agree with the referee. The $\Delta relA ytfK$ double knock-out strain produces significantly less ppGpp level than $\Delta ytfK$ knock-out strain (see figure 3B of the revised manuscript right panel). The results in Figure 3B and 4A of the revised manuscript, only argue that the ppGpp level observed in the $\Delta ytfK$ mutant is sufficient to temporarily survive cerulenin treatment.

e) Note that the relationships between cerulenin, (p)ppGpp and ampicillin tolerance have been extensively investigated by the lab of Edward Ishiguro, who concluded using ppGpp0 strains that ‘Penicillin tolerance was shown to be a direct consequence of the inhibition of phospholipid synthesis and not due to the possible accumulation of guanosine-3’,5’-bispyrophosphate (ppGpp), the starvation stress signal molecule known to be responsible for the development of penicillin tolerance in amino-acid- deprived bacteria.’ (Rodionov and Ishiguro 1996). No references to Ishiguro’s works in the paper by Germain and colleagues. I suggest amending this, and since there are clear differences in ppGpp0 behaviour upon cerulenin challenge in Ishiguro’s lab and in the current paper, it is prudent to check if the strain used in the current work is not contaminated by phages (see works by Maisonneuve and colleagues recently retracted from PNAS and Cell)

AU: We thank the referee for embarrassing us. As shown below diagnostic PCR confirm the absence of prophages for the five strains used in the manuscript (wt, $\Delta relA$, $\Delta ytfK$, $\Delta relA ytfK$, $\Delta relA spoT$). (Note that the $\Delta relA spoT$ used in the present manuscript is different from the strain which is infected by Phi80 in the Maisonneuve et al. 2013 paper¹¹. Indeed we newly transduced the $\Delta relA::251 kan$ allele and the $\Delta spoT::207 cat$ allele in the MG1655 and confirm (see below) absence of phi80 and Lambda lysogenization.

Diagnostic multiplexed PCR on strains described in manuscript.

wt, $\Delta relA$, $\Delta ytfK$, $\Delta relA ytfK$, $\Delta relA::251Kan spoT::Cat$, $\Delta relA::251Kan$ and the infected $\Delta 10^{11}$ strain with phi80 and lambda were tested by diagnostic multiplex PCR using primers described in Harms et al., 2017¹². Briefly the *yciI* is the WT locus where Phi80 integrates. *kch-int* and *pinL-tonB* loci are amplified when Phi80 phage is lysogenized in the chromosome at *yciI* locus (Left panel). *gal-bio* is the WT loci while *b2-bio* is amplified when lambda phage is lysogenized in the chromosome at *attB* attachment site (Right panel).

We apologized for omitting to correctly cite the work by Ishiguro and colleague showing that fatty acid starvation induced penicillin tolerance. The reference has been added accordingly in the revised manuscript. However the referee comment about the different behaviors of ppGpp0 strain regarding penicillin (ampicillin) tolerance after cerulenin treatment is irrelevant:

1- Simply because we did not perform ampicillin treatment in $\Delta relA spoT$ mutant after cerulenin treatment (as explained below).

2- The experimental set up present in Rodionov and Ishiguro work is fundamentally different from our experiments¹³. They followed the optical density as function of time in presence of ampicillin with and without cerulenin. The conclusion that such experiment provides is that indeed ppGpp is not required for ampicillin induced lysis. However, in light of results obtained by Petra Levin's group^{14,15} and our own results, we wonder, how cells of a ppGpp0 strain could tolerate ampicillin treatment if they are already killed by the addition of Cerulenin (Figure 4A) ?

f) Line 160 'These results strongly suggest that, indeed, elevated (p)ppGpp level confers antibiotic tolerance' I feel that the data are not strong enough to provide the evidence for this statement. The inhibition of FAS by cerulenin treatment renders cells non-growing due to depletion of phospholipid precursors, this should not be dependent on (p)ppGpp production. This in its self should confer tolerance to ampicillin? (p)ppGpp is produced in response to the stress and aids adaptation of the cell and cell survival, i. e. can you say from this data that this is the driving force that confers antibiotic tolerance? Again, growth curves are essential.

AU: we agree with the referee that the sentence was overstated and we corrected it accordingly. However our results clearly show that (p)ppGpp contributes to ampicillin tolerance.

10. General comment on data presentation: The authors use various constructs in various many experiments. It is essential that it is clear which strains and plasmids are used where, this needs to be unambiguously specified in figure legends with a reference to plasmid and strain table. Similarly, different concentrations of IPTG are used in various figures: 1 mM on Figure 1, 500 uM on Figure 4 and 200 uM on Figure 3S. This unnecessarily complicates direct comparisons.

AU: We apologize for the lack of clarity and organization in the presentation of our results at the first stage of submission. For the sake of clarity we are now providing results obtained with all tested concentrations of inducer in supplementary figures.

To conclude: starting from the Abstract, the authors rename YtfK to TspT, Trigger of SpoT (and the abbreviation TspT is first spelled out on line 110). At this moment the results are not solid enough: Trigger of SpoT AND RelA is a possible interpretation as well. I feel that trademarking the data is premature at this point.

AU: We took this remark in consideration. Despite strong arguments supporting the role of YtfK on SpoT activities, we cannot totally rule out that YtfK does not activate RelA, therefore we decided not to rename YtfK and we change the discussion accordingly.

Reviewer #2 (Remarks to the Author):

In this very important work, the authors report the discovery of a new factor, TspT, required for the ppGpp synthase activity of SpoT in *Escherichia coli*. The findings are compelling, since in a *tspT* mutant, there is no ppGpp apparition in stress conditions known to normally trigger SpoT-dependent ppGpp accumulation. And in reverse, the overproduction of TspT is enough to trigger ppGpp increase. This finding is of primary importance in the field, given the role of ppGpp in survival and antibiotics resistance, and given that we know so little about the mechanism of control of the enzymes of the SpoT family.

AU: We thank the Reviewer for his/her very positive attitude and for helpful comments

We address the questions below.

General comments

The experiments clearly demonstrate the requirement of TspT for SpoT synthase activity. However, it is not clear if the role of TspT is really to detect starvation and play a role in the regulation of SpoT per se, or if simply TspT presence is needed for SpoT synthase activity. The authors show the effect of TspT in conditions of phosphate starvation and fatty acid starvation, but it is very easy to imagine that the same effect might be observed for any conditions triggering ppGpp synthesis in the absence of RelA.

AU: The referee is absolutely correct and we are currently testing this possibility. However in absence of compilation data on the role of YtfK in all different conditions known to trigger SpoT we prefer not to discuss this possibility in the present context.

If there is really a regulation of SpoT activity by TspT, then it is likely to be acting by a change in TspT levels (suggested by the activation obtained simply by TspT overproduction). There are published data about the regulation of *tspT* expression by the PhoB transcriptional activator. Does the observations reported here correlated with changes in the amounts of TspT protein? What are the ratio of TspT and SpoT proteins in the cell in balanced conditions or during stress response ?

Such information is important to interpret the kinetic constant measured for the TspT/SpoT interaction in vitro. It would also be interesting to get an idea of the levels of expression (*tspT* gene expression or TspT protein levels) in the different conditions tested. Also, the results of the 2017 paper on TspT should be discussed in regard of the results of the present work.

AU: We thanks the referee for this very important question. To address this point, we have constructed a translational fusion of YtfK with a green fluorescent reporter and followed the protein expression level of YtfK during stress conditions (Phosphate, Fatty acid, and Amino acid starvation). As shown in Figure S3, *ytfK* is not only induced during phosphate starvation (Figure S3A) but also during fatty acid starvation (Figure S3B). Interestingly, YtfK level is not

affected during amino acid starvation strongly arguing on the absence of physiological role of YtfK during this condition and then supporting the specific activity of YtfK on SpoT activities.

We agree with the referee that ectopic production of YtfK triggers SpoT-dependent (p)ppGpp accumulation in absence of external stresses and therefore raises the possibility that the regulation of SpoT activity by YtfK is likely to be acting by a change in the SpoT/YtfK ratio. We therefore engineered a genetic tools to test this assumption using growth on SMG plates as a readout for (p)ppGpp level. As observed in Figure 2 and S2 of the revised manuscript, the SpoT-YtfK ratio controls the switching from (p)ppGpp degradation to synthesis. Moreover we observed that SpoT synthetase activity is subjected to YtfK limitation *in vivo*.

It would be important to give rigorous information about the conservation of tspT in bacteria. Is TspT systematically present in bacteria having two distinct RSH enzymes SpoT and RelA ? Is it present in other types of bacteria ?

AU: YtfK is specific to Gamma-proteobacteria that seems limited to orders of *Enterobacteriales*, *Vibrionales*, *Aeromonadales*, *Pasteurellales* and *Alteromonadales*¹. This information is now provided in the discussion of the revised manuscript.

Specific comments

In the discussion, the authors show a figure of a 3D modelling of the potential TspT-SpoT complex, based on the structure of RelSeq enzyme and a structure of TspT available in the pdb. However, there is very little information on how the model was obtained (parameters, assessment of the quality of the final model for example), and is it not necessary to have access to the structure file? Furthermore, there is very little use of this model in the paper, for example the position of the Proline 42 of TspT shown to be important for the interaction with SpoT is not even shown on the figure or discussed.

AU: We agree with the referee. The model is now presented in supplementary information Figure S8. We apologize for the lack of sufficient description in our manuscript at the first stage of submission. We modified the material and methods accordingly in the supplementary information.

Concerning the 2hybrid : the results are overall very clear and convincing. However, it is important to show the correct expression of the diverse hybrid proteins (especially for constructs that do not interact). In addition, the results about specificity of interaction (no interaction with RelA) are shown in only one combination of vectors. Same for the identification of the domain of interaction. The authors should at least explain why they choose a combination rather than the other.

AU: We thanks the referee for this important remark. We have confirmed the lack of interaction between RelA and YtfK in the 8 tested combination/orientations (Figure S5B of the revised

manuscript). Moreover RelA multimerizes *in vivo*⁸ and we used this readout to confirm the correct production of the RelA protein fusion from both plasmids (Figure S5D).

Regarding the choose of the combinations used for BTH; given that the SpoT-YtfK interaction was observed in both combination of plasmids (Figure S5A) and given that SpoT is not abundant *in vivo* it appeared physiologically relevant to pursue the study with SpoT in the pKT25 plasmid (which is a low copy number plasmid).

Finally, we agree with the reviewer that following the expression level of the truncated version of SpoT would have been a good control. We tried it but unfortunately, commercial available antibodies against CyA (santa Cruz Ref sc-13582 (9D4) and Ref SC-13581 (3D1)), were not reactive against the T25 fragment. Finally, we confirm the correct production of the pKT25-spoT-HD and pKT25-Cter-spoT because they interact with a new partner of SpoT which is currently under characterization in the laboratory. If required, we can provide these information at **the discretion of the editor and the referee 2**.

Concerning the *in vitro* binding assay : because the K_d obtained is rather low, the YtfK(P42L) mutant, and importantly the catalytic domain of RelA should be used as controls.

AU: Unfortunately the YtfK P42L protein is unstable *in vitro* and rapidly aggregate during thioredoxin-tagged cleavage and cannot be used *in vitro*. Moreover, we agree on the point that RelA might be a good control for *in vitro* experiments. However, long RSHs are very challenging protein for biochemistry and tends to precipitate. Therefore, in absence of interaction between YtfK and RelA *in vivo* and in absence of tools in hands to confirm RelA activity *in vitro*, we prefer to abstain to address this point.

Finally, we agree that the apparent K_D obtained *in vitro* by Biolayer interferometry suggested indeed a weak and transient/ highly dynamic interaction (Figure 7A of the revised manuscript). We believe that the question is rather to understand if this interaction is physiologically productive (ie specific). Our *in vivo* experiments show that SpoT-YtfK interaction seems specific to SpoT and functionally relevant.

Finally we provide additional biochemical experiments supporting the rational for the YtfK-SpoT interaction *in vitro*. Dynamic light scattering approaches enable us to confirm this interaction and provide us with new evidences for SpoT conformational changes upon YtfK binding as shown in Figure 7 of the revised manuscript

Minor comments

The strain constructions might be described a bit more (how the kanaR cassette was removed ? describe the two successive P1 transduction for ppGpp^o strain construction for example etc...).

AU: We apologize for this omission, and we have changed the supplemental materials and methods accordingly.

There is a very lengthy description of the methods for protein production and purification, while other equally important parts are overly succinctly described or missing (initial screening for tspT and mutagenesis screen on tspT; 3D modelling; strain construction). For example, for the screen using the ASKA collection, the description of the expression clones is important : are the genes tagged, with 6his or with GFP ?

AU: We apologize for these omissions. We improved the text accordingly.

For the screen, we used the ASKA collection without GFP tag but with a 6His tag in the N-terminal domain of the protein of interest. Immediately after screening and sequencing, we properly cloned, untagged version of each candidate in the pEG25 plasmid (see Methods section). We improved Method sections text accordingly.

figure S2 : what is the meaning of -ve/+ve ? If it means “empty vector”, why is it sometimes negative, sometimes positive? What is the difference between the left panels and the right panels ? They seem completely redundant.

AU: The referee is correct. We apologized for the typo mistakes in the Figure S2. Changes have been made accordingly in the revised Figure S5.

Reviewer #3 (Remarks to the Author):

The stringent response is a conserved regulatory mechanism allowing bacteria to adapt to a variety of stressful conditions. Recently, major new insights were gained into the molecular mechanisms. Here the author made a further major contribution showing that the enzymatic activity of the bifunctional enzyme SpoT is modulated through interaction with a small cytoplasmic protein, here named TspT. This is an interesting finding which shed new lights onto the long-lasting question how SpoT contributes to stringent response in *E. coli*. The manuscript is well written. However at some points a more thorough analysis could help to get a more complete picture. Some questions can be addressed with the tools in hands. e.g. How is Tsp regulated under fatty acid starving conditions? Does Tsp interaction with SpoT contribute to H2O2 tolerance? How does TspT influence the synthetase/hydrolase activity of SpoT *in vitro*?

AU: We thank the Reviewer for his/her very positive attitude and for helpful comments.

We address the questions below.

Specific comments

Line 28 and throughout: Since the identified SpoT interacting protein already has a designation (YtfK) and was already shown to be regulated by Pho as well as to be involved in oxidative stress survival I suggest to stick to YtfK instead of renaming it to TspT.

AU: When the *E. coli* genome sequence was annotated, genes encoding for protein of unknown function were specified as “y” gene. In the case of YtfK we believe that “y” designations is suboptimal. However given similar remark from reviewer 1 and given that the direct molecular function of YtfK is not fully demonstrated *in vitro* we agree that renaming YtfK is premature at this point. Changes have been done accordingly.

Figure 2C: As it seems YtfK/SpoT induces only ppGpp not pppGpp. This should be mentioned and discussed.

AU: It is indeed an interesting/intriguing observation and as mentioned below we do not yet know the exact reason for that. Therefore we prefer to abstain from providing general statements on this observation.

Previous kinetic labelling experiments with ^3H uracil in ^{32}P labeled cells followed the label from GTP to (p)ppGpp. During amino acid starvation ^3H first appeared in GTP and pppGpp and rapidly went to ppGpp. GTP is the most abundant in the pool after AA starved so ppGpp is formed via the Guanosine 5'-triphosphate, 3'-pyrophosphatase (GppA) that catalyzes the conversion of pppGpp to ppGpp.

During fatty acid or phosphate starvation, the kinetics are much slower and GppA may have sufficient activity to totally convert pppGpp to ppGpp. However our preliminary results (see below) suggest that it is not the case as no detectable pppGpp is observed in a GppA deletion strain during fatty acid starvation. Alternative pathway (ie exopolyphosphatase ppx) might also contribute to pppGpp conversion

There is also the good possibility that GDP becomes a more abundant or a preferred substrate during these stresses.

[Redacted]

GppA does not affect the ppGpp/pppGpp ration under fatty acid starvation. Exponentially growing cells of MG1655 (WT) and isogenic deletion strains, *ΔrelA*, *ΔgppA*, were challenged for fatty acid starvation by addition of 250μg/mL of cerulenin. After 30 min treatment nucleotides were extracted and separated by TLC. Representative autoradiograph of the TLC is shown.

Line 117, Figure 2A: Because the link of phosphate starvation and ppGpp (via TSP) is central to the whole story, quantitate results based on the three independent biological replicates should be shown.

AU: We now provided mean quantification and standard deviation for ppGpp accumulation observed during the three starvation conditions but also for experiments with overproduction of YtfK and YtfK^{P42L} (Figure 3 and 5B-C respectively).

Line 155, Figure 2 B: The GTP spot as indicated is confusing. Why GTP is only visible after Cerulenin addition?

AU: as seen in the uncropped TLC below the GTP is present but migrates, for an unknown reason slightly higher than the GTP spot observed in samples treated with cerulenin.

Line 145 and following, Figure 3: It should be indicated whether there are any differences in growth or MIC (cerulenin, ampicillin) between the strains analysed? For results shown in figure 3 B and 3 C the $relA/spoT$ and $relA/tsp$ mutant should be included in the analyses, to show that the effects are due to TSP mediated ppGpp synthesis. This mutant should have the same or at least very similar phenotype to the tsp mutant. It would be also helpful to know whether TspT expression is increased by cerulenin. YtfK/Tsp was already shown to be involved in tolerance towards H₂O₂ and phosphate starvation (Iwadate Y, Kato JI. Microbiology. 2017;163:1912). It would be very interesting to see whether this also mediated via SpoT.

AU: We thank the referee for this important remark. We have performed additional experiments and we now provided growth curves of WT, $\Delta relA$, $\Delta ytfK$, $\Delta relA ytfK$, and $\Delta relA spoT$ with and without cerulenin treatment (Figure 4 and S4 of the revised manuscript).

The Figure 4A and 4B show that cerulenin becomes bactericidal in $\Delta relA ytfK$ and $\Delta relA spoT$ mutant. Given that these mutants cannot survive cerulenin treatment rendered the use of these mutants in a combination assay (cerulenin and ampicillin) not relevant.

Regarding the expression of *ytfK*, we constructed a translational fusion of YtfK with green fluorescent reporter. As shown in Figure S3 of the revised manuscript, *ytfK* expression is triggered both by fatty acid and phosphate starvation but not during amino acid starvation strongly arguing on the specific role of YtfK on SpoT.

Regarding H₂O₂ tolerance, while stringent response and oxidative stress are linked¹⁶ a direct role of SpoT is not yet established. Interestingly the data presented by Iwadate and colleague show that *ytfK* contributes to H₂O₂ tolerance by stimulating expression of at least the catalase

encoded by *katG*. Whether the stimulation is direct or indirect through activation of SpoT is an interesting question and have decided to discuss it in the revised manuscript.

Note that *E. coli* biofilm formed by *ArelA spoT* mutant are defective for catalase activity¹⁷ pointing to an important role of ppGpp in regulation of the catalase (at transcriptional and post translational level). Moreover ppGpp activate the binding of RNA polymerase to the *katE* promoter. However the results presented Iwate et al point to the requirement of the constitutive catalase KatG¹⁰ suggesting a more complex regulation.

Line 10: Since the authors already have purified Tsp and SpoT in hand it should be feasible to perform an in vitro activity assay to finally confirm that Tsp activates ppGpp synthesis or possibly inhibits ppGpp degradation.

AU: We agree with the referee that such experiment will be extremely useful. It might be realized that such enzymatic *in vitro* reconstitution are much harder than it looks and may require full length SpoT as well as point mutant affecting either hydrolysis or synthesis in order to conclude. Therefore it is our sincere opinion that such results are not required for our story to be complete and might be more appropriate in an independent story on the molecular/structural mechanisms behind the regulation of SpoT by YtfK.

Discussion:

Some more information on YtfK/TspT should be given. Is it conserved in different bacteria? Localisation, basic protein? It was recently shown that YtfK is involved in H₂O₂ tolerance (Iwate Y and Kato J, Microbiology). This should be at least discussed. Is there anything known about the regulation of ytfK besides being part of the Pho regulon.

Line 340: What is meant with suppress or complement the growth defect? You probably screened for growth

AU: We apologize for this omission. These important information on YtfK have been introduced in the first paragraph of the discussion section. The work by Iwate and colleague is now further discussed (see above).

We indeed screen for growth. Clarification has been made.

References

1. Gao, B., Mohan, R. & Gupta, R. S. Phylogenomics and protein signatures elucidating the evolutionary relationships among the Gammaproteobacteria. *Int. J. Syst. Evol. Microbiol.* (2009). doi:10.1099/ijs.0.002741-0
2. Ringquist, S. *et al.* Translation initiation in Escherichia coli: sequences within the ribosome-binding site. *Mol. Microbiol.* (1992). doi:10.1111/j.1365-2958.1992.tb01561.x
3. Xiao, H. *et al.* Residual guanosine 3',5'-bispyrophosphate synthetic activity of relA null mutants can be eliminated by spoT null mutations. *J. Biol. Chem.* (1991).

4. Mechold, U., Murphy, H., Brown, L. & Cashel, M. Intramolecular regulation of the opposing (p)ppGpp catalytic activities of RelSeq, the Rel/Spo enzyme from *Streptococcus equisimilis*. *J. Bacteriol.* (2002). doi:10.1128/JB.184.11.2878-2888.2002
5. Sinha, A. K., Winther, K. S., Roghanian, M. & Gerdes, K. Fatty acid starvation activates RelA by depleting lysine precursor pyruvate. *bioRxiv* 635748 (2019). doi:10.1101/635748
6. Primakoff, P. & Artz, S. W. Positive control of lac operon expression in vitro by guanosine 5'-diphosphate 3'-diphosphate. *Proc. Natl. Acad. Sci.* (2006). doi:10.1073/pnas.76.4.1726
7. Battesti, A. & Bouveret, E. The bacterial two-hybrid system based on adenylate cyclase reconstitution in *Escherichia coli*. *Methods* (2012). doi:10.1016/j.ymeth.2012.07.018
8. Gropp, M., Strausz, Y., Gross, M. & Glaser, G. Regulation of *Escherichia coli* RelA requires oligomerization of the C-terminal domain. *J. Bacteriol.* (2001). doi:10.1128/JB.183.2.570-579.2001
9. Kudrin, P. *et al.* The ribosomal A-site finger is crucial for binding and activation of the stringent factor RelA. *Nucleic Acids Res.* (2018). doi:10.1093/nar/gky023
10. Iwadate, Y. & Kato, J. I. Involvement of the ytfK gene from the PHoB regulon in stationary-phase H₂O₂ stress tolerance in *Escherichia coli*. *Microbiol. (United Kingdom)* (2017). doi:10.1099/mic.0.000534
11. Maisonneuve, E., Castro-Camargo, M. & Gerdes, K. (p)ppGpp controls bacterial persistence by stochastic induction of toxin-antitoxin activity. *Cell* **154**, (2013).
12. Harms, A., Fino, C., Sørensen, M. A., Semsey, S. & Gerdes, K. Prophages and growth dynamics confound experimental results with antibiotic-tolerant persister cells. *MBio* (2017). doi:10.1128/mBio.01964-17
13. Rodionov, D. G. & Ishiguro, E. E. Dependence of peptidoglycan metabolism on phospholipid synthesis during growth of *Escherichia coli*. *Microbiology* (1996). doi:10.1099/13500872-142-10-2871
14. Vadia, S. *et al.* Fatty Acid Availability Sets Cell Envelope Capacity and Dictates Microbial Cell Size. *Curr. Biol.* **27**, 1757-1767.e5 (2017).
15. Westfall, C. *et al.* The widely used antimicrobial triclosan induces high levels of antibiotic tolerance in vitro and reduces antibiotic efficacy up to 100-fold in vivo. *Antimicrob. Agents Chemother.* (2019). doi:10.1128/aac.02312-18
16. Chang, D. E., Smalley, D. J. & Conway, T. Gene expression profiling of *Escherichia coli* growth transitions: An expanded stringent response model. *Mol. Microbiol.* (2002). doi:10.1046/j.1365-2958.2002.03001.x
17. Nguyen, D. *et al.* Active starvation responses mediate antibiotic tolerance in biofilms and nutrient-limited bacteria. *Science* (80-.). (2011). doi:10.1126/science.1211037

Reviewers' comments:

Reviewer #1 (Remarks to the Author):

In the revised version of the paper by Germain and colleagues the microbiology part of the paper has improved: some of the inconsistencies have been ironed out and TLC data were quantified. However, some of the crucial issues have not been resolved. In fact, my initial concerns were reaffirmed. The most crucial is the lack of biochemical validation, which has now become even more essential than during the original submission.

- i) the DLS experiments added to the revised version do not support specific stoichiometric complex formation between SpoT and YtfK but rather suggest non-specific aggregation, thus invalidating the binding studies and rendering the microbiological experiments relying on YtfK overexpression questionable
- ii) the YtfKP42L mutant seems structurally destabilised rather than unable to specifically bind SpoT
- iii) new data on BioRxiv from Prof. Gerdes (a co-author on the current submission) suggests that cerulenin used in this study to induce fatty acid starvation promotes ppGpp accumulation mediated by RelA, instead of specifically triggering SpoT.

Below I motivate my concerns in detail. In the current form I would recommend rejection of the manuscript.

Growth inhibition by YtfK: Figure 5D, 'The intracellular level of (p)ppGpp induced by ytfK overexpression controls the growth rate' shows that expression of YtfK inhibits growth in the wt background, and, less efficiently, in the relA KO background. At the same time, no growth inhibition is observed on Figure 1 (relA KO) and 2 (wt). I guess the reason is the different concentrations of IPTG – 200 μ M on Figure 1, up to 75 μ M on Figure 2 and 1 mM on Figure 5D – the authors should comment on this in the text, otherwise it might be confusing. Note that SpoT expression levels are never quantified directly.

Importantly, Figure 5D lacks the essential control for connecting YtfK-mediated growth inhibition to ppGpp accumulation: an experiment in the ppGpp⁰ background. Note that while RelA does not interact with YtfK in the bacterial two hybrid assays there is a RelA-dependent component in ppGpp accumulation driven by YtfK overexpression – the authors should comment on this contradiction in the manuscript. The YtfKP42L mutation seems not to specifically disrupt the interaction with SpoT, but rather acts via destabilisation of the structure of YtfK: attempts to purify the protein failed, (which should be mentioned in the manuscript), and therefore cannot be used as a sufficient specificity control.

Due to the above-mentioned experimental issues it is absolutely essential to run the enzymatic assays suggested during the first round of revision. I am not satisfied with the response of the authors who despite a direct request from me and the second reviewer, do not perform this simple experiment: just saying 'It is our sincere opinion that such results are not required for our story' is not a sufficiently strong argument, especially given the alarming DLS results (see below). The authors have already purified SpoT and YtfK, and RelA purification well-documented and is in no way more challenging than that of SpoT and YtfK. The TLC assay is set up, and all that is needed is adding the substrates and resolving the products. It literally takes one day to test the effect of YtfK on SpoT. Testing the effect on RelA would take a week.

I find it very strange that while the authors performed multiple additional in vitro experiments (DLS), but decline performing a simple, direct, crucial enzymatic test. As far as I am concerned, this remains an absolutely essential experiment – especially given that now we all agree that the current data can be interpreted that YtfK activates RelA as well.

Biochemical experiments became especially essential given that the authors themselves (Sinha et

al. 'Fatty acid starvation activates RelA by depleting lysine precursor pyruvate') have recently shown that a cerulenin challenge causes amino acid starvation leading to RelA-dependent accumulation of ppGpp (note that the authors extensively use cerulenin in their experiments, Figures 3 and 4). Note also that Sinha et al. (<https://www.biorxiv.org/content/10.1101/635748v1>) is not cited in the paper. It absolutely should be discussed since this result is crucial for interpreting the experiments. Here is a quote from the abstract: 'Here, we discover that FA starvation leads to rapid activation of RelA and reveal the underlying mechanism. We show that fatty acid starvation leads to depletion of lysine that, in turn, leads to the accumulation of uncharged tRNA_{Lys} and activation of RelA. SpoT was also activated by fatty acid starvation but to a lower level and with a delayed kinetics.'

Figure 4: I am unsure that the results on tolerance are warranted in the main text and in the abstract: the loss of YtfK does not have a strong phenotype under clinically-relevant antibiotic challenges (e.g. ampicillin challenge), and there is an effect only with combinatorial knock outs (YtfK and RelA) or clinically-irrelevant combinatorial antibiotic challenges (ampicillin + cerulenin). Stating that 'YtfK contributes to antibiotic tolerance' is borderline misleading: while the authors can, indeed, find a clinically irrelevant condition generating a phenotype, this does not advance our understanding of biology. I again strongly recommend moving this figure to the supplement and removing the statement from the abstract: there is no point over-heating the ppGpp vs antibiotic tolerance field.

The next point is related to the role of ppGpp in antibiotic tolerance. It would be judicious for the authors to remove the citation to their retracted paper on the topic: 34. Germain, E., Roghanian, M., Gerdes, K. & Maisonneuve, E. Stochastic induction of persister cells by HipA through (p)ppGpp-mediated activation of mRNA endonucleases. *Proc. Natl. Acad. Sci. U. S. A.* 112, (2015).

The newly added DLS experiments show clearly that SpoT does not form a 1:1 complex with YtfK. YtfK itself gives a sharp peak, and if a 1:1 complex would form, the SpoT peak would shift. Instead it becomes more diffuse and heavy-tailed, which strongly speaks against formation of a specific 1:1 complex. What is alarming is in all of the experiments performed in the presence of increasing concentrations of YtfK added to SpoT, there is no separate YtfK peak. This is inconsistent with 1:1 complex formation: at the highest concentration of YtfK (120 μ M), there is a 90 μ M YtfK excess over 30 μ M SpoT. These 90 μ M of YtfK should give a peak – and there is none. Instead of several well-defined mass peaks corresponding to free YtfK, free SpoT (if 20 μ M Kd is correct, there should always be free SpoT) and to YtfK:SpoT peak (poorly resolved with SpoT) there is a broad distribution with a heavy tail, and the tail becomes heavier as YtfK is added. This is all indicative of proteins sticking together non-stoichiometrically, and in the absence of specific controls and enzymatic assays, this invalidates not only the in vitro interactions studies by interferometry – but also suggests that similar non-specific effects are happening in microbiological assays relying on overexpression, thus compromising the rest of the study.

To summarise: proper biochemical experiments supported by DLS experiments demonstrating specific 1:1 complex formation between YtfK and SpoT are absolutely essential. Right now the additional experiments only reinforced my concerns.

Minor:

Abstract and Introduction: "synthetase (SYNTH) and hydrolase (HD) motifs" – SYNTH and HD are not motifs. These are either protein domains that perform an enzymatic activity, or in the case of HD a small sequence motif present in metal-dependent phosphohydrolases (please see Aravin and Koonin 1998). The C-terminal half of RSHs proteins is not a domain either – it is a region encompasses several domains. The definition of a domain in structural biology is quite clear: it is part of a given protein that can evolve, function, and fold independently of the rest of the polypeptide. The authors should revise the manuscript throughout in accordance to established concepts and definitions. The statement in the introduction "N-terminal domain of SpoT

encompassing the hydrolase and synthetase domains" epitomizes these issues.

Introduction, line 56-57: 'SpoT has both (p)ppGpp hydrolytic and synthetic activities, similar to RSHs10–12.' Something is wrong here: since SpoT is an RSH, it cannot be 'similar to RSHs'. I guess the authors meant Rel, long bi-functional RSH enzyme present in the majority of bacterial species.

BLI experiments: The data was analysed using steady state model, however the sensograms collected for the higher concentration values do not reach steady state (required to report an equilibrium dissociation constant) and it seems that 120 (... 120 seconds? note that the X-axis is not labeled in the figure; it should be) is not enough. Not reaching a clear equilibrium is indicative of experimental setup problems (see above; DLS results). In addition the material methods for this part is lacking, e.g. it is not clear as to how the data was treated to account for the baseline drift and why was the experiment stopped after 90 s.

Docking: It is not clear how was the docking performed: which program was used? what protein was used as receptor and which as ligand? What score criteria was used for selection? Docking programs do not give a single model, what were the five best model picked up by the docking model and why was this one selected over the others? All these issues should be addressed in the material and methods to be able to reproduce the docking experiment and extract a meaningful insight from it, other this should be removed altogether from the paper.

When presenting the data as a scatter plot (e.g. Figure 3), polynomial fitting in Excel should be avoided – use 'line and markers' representation instead. When you fit the data, you subscribe to a certain model that describes the underlying process, be it Hill, exponential decay, Michaelis-Menten. Clearly, the kinetics of ppGpp accumulation is not governed by polynomial equations – and the only function the trace has in this case is a guide for the eye. Connecting the markers with straight lines does exactly that.

Protein purification: when working with RNA-binding proteins, it is crucial to report the 260/280 ratio for the final preparations. Without these, it is impossible to assess the quality of the preparations. If either SpoT or YtfK comes with RNA, and the other protein has affinity to RNA, this could explain the nonspecific effects observed by DLS.

Line 349: *Streptococcus equisimilis* – no need to capitalise 'equisimilis'

Reviewer #2 (Remarks to the Author):

General comments

In this revised manuscript, the authors have strengthened their findings that YtfK is an essential factor for the functioning of SpoT in response to phosphate starvation and cerulenin treatment. In particular, they provide new important experiments showing that ytfK is induced in the stress conditions tested. Furthermore, by controlling the level of expression of SpoT and YtfK in vivo, they show that the ratio between the 2 proteins is likely to play a role in SpoT enzymatic regulation.

They have made a good job in responding to most of the reviewers comments. They simply argue that in vitro assay of enzymatic activity is too much to ask at this point given the difficulty to work on these proteins.

Concerning the two-hybrid, if it is true that the T25 fusions cannot be detected easily, the sc13582 antibody is very good to detect the T18 fusions. It might have been possible to show the correct

production of all the T18 hybrid proteins in general, an especially to compare the T18-YtfK(P42L) and T18-YtfK fusions, because as stated by the authors, there might be a problem of stability of the mutant construct. For the in vitro binding study, a control with the Nterminal domain of RelA would still have been an additional strength, maybe with a point mutant in the synthesis domain to help for the expression.

It might be interesting to discuss why the ytfK mutant is not impaired in survival to cerulenin treatment, while it has an effect in stress-induced ampicillin tolerance. This later experiment is in fact the combination of 2 antibiotic treatments, and it might have been more clear to test the ampicillin tolerance with a stress induced by starvation (i.e. phosphate for example).

Minor comments:

Line 46 : there are more recent articles by the Gourse's lab than the reference given here, describing the 2 binding sites of ppGpp on RNAP.

Lines 96 and 99 : I think I understand that the new cloning in pEG25 was to remove the tag, but the reader cannot : explain why a new cloning.

Lines 178-180 : I am confused by the numbers 1,000 fol and 10,000 fold . From what I understand from the figure, the ppGpp^o strain is more affected than the relA_{ytfK} mutant.

Line 268 : typo : oligomerization OF YtfK

Line 276 : words missing ? Catalytic « domain presents » conformation changes ?

Lines 294-298 : rather than showing that YtfK is involved in antibiotic tolerance, I think that the results shown here are simply « consistent » with the role of YtfK in controlling SpoT and ppGpp levels. And then as a consequence the antibio tolerance because it is already known that ppGpp levels are involved in this phenomenon. I would therefore not say that the results add anything new to the role of SpoT and ppGpp in the control of bacterial multi drug tolerance.

Line 340 : typo : RSH

Reviewer #3 (Remarks to the Author):

In the revised manuscript most of the previous concern are addressed or discussed.

We thank the reviewers for critically reviewing the revised manuscript. The comments have helped us to improve the manuscript. We have accommodated most changes and added experiments suggested by the referees. Text modifications are appearing in blue in the core body of the revised manuscript.

Below we have explained point-by-point how we have addressed your questions and concerns to strengthen our manuscript.

Kind regards

Reviewer #1 (Remarks to the Author):

In the revised version of the paper by Germain and colleagues the microbiology part of the paper has improved: some of the inconsistencies have been ironed out and TLC data were quantified. However, some of the crucial issues have not been resolved. In fact, my initial concerns were reaffirmed. The most crucial is the lack of biochemical validation, which has now become even more essential than during the original submission.

i) the DLS experiments added to the revised version do not support specific stoichiometric complex formation between SpoT and YtfK but rather suggest non-specific aggregation, thus invalidating the binding studies and rendering the microbiological experiments relying on YtfK overexpression questionable

We disagree with the reviewers' remark as regarding to non-specific aggregation between SpoT¹⁻³⁷⁸ and YtfK. We used the Z-average parameter, which is the most reliable value in DLS; it is used as a quality control setting defined in ISO 22412. Z-average is defined as the mean particle size by intensity. This value is highly sensitive to particle aggregation. We have reported the Z-average values for SpoT¹⁻³⁷⁸ alone and in solution with YtfK in Table S1. It should be noted that SpoT¹⁻³⁷⁸ alone has a Z-average of 11.39±0.13nm for five replicates and when incubated with YtfK has a Z-average of 11.81±0.19nm from 19 measurements. We now also displayed the 90% percentile size Dv(90) for all experiments in the supplementary Table S1 of the revised manuscript. All samples have sizes less than 13nm for 90% of the population. Raw correlation data before integration is now provided in the revised Figure 7B. Taken into consideration these data, there is no non-specific aggregation when SpoT¹⁻³⁷⁸ is interacting with YtfK.

ii) the YtfKP42L mutant seems structurally destabilised rather than unable to specifically bind SpoT

AU: see response below

iii) new data on BioRxiv from Prof. Gerdes (a co-author on the current submission) suggests that cerulenin used in this study to induce fatty acid starvation promotes ppGpp accumulation mediated by RelA, instead of specifically triggering SpoT.

AU: see response below

Below I motivate my concerns in detail. In the current form I would recommend rejection of the manuscript.

AU: The severe criticism from Reviewer 1 confuses us. We hope that the additional experiments, clarifications, and improvements of the text will be sufficient to convince Reviewer 1 that our work is sound and robust.

Growth inhibition by YtfK: Figure 5D, ‘The intracellular level of (p)ppGpp induced by ytfK overexpression controls the growth rate’ shows that expression of YtfK inhibits growth in the wt background, and, less efficiently, in the relA KO background. At the same time, no growth inhibition is observed on Figure 1 (relA KO) and 2 (wt). I guess the reason is the different concentrations of IPTG – 200 uM on Figure 1, up to 75 uM on Figure 2 and 1 mM on Figure 5D – the authors should comment on this in the text, otherwise it might be confusing. Note that SpoT expression levels are never quantified directly.

AU: we do not really understand the referee concern. The concentration are clearly written in the text, the figures and the figure legends. Importantly the results presented in Figure 5D (and supplementary Figure 6) of the revised manuscript show that at high concentration of IPTG in liquid media YtfK reduces the growth rate (which is different from a growth arrest) in a (p)ppGpp dependent manner. The Figure 1 and 2 present a different assay done on plates (growth VS no growth) where the growth rate is nearly impossible to assess since the reading is performed after 36h hours of incubation at 37 °C. Therefore we do not believe that such comparison is appropriate.

Importantly, Figure 5D lacks the essential control for connecting YtfK-mediated growth inhibition to ppGpp accumulation: an experiment in the ppGpp⁰ background.

AU: We agree with the referee remark. We performed additional controls showing that indeed, overexpression of YtfK does not affect the growth rate of a ppGpp⁰ strain. The results are shown in the supplementary Figure 6 of the revised manuscript.

Note that while RelA does not interact with YtfK in the bacterial two hybrid assays there is a RelA-dependent component in ppGpp accumulation driven by YtfK overexpression – the authors should comment on this contradiction in the manuscript. The YtfKP42L mutation seems not to specifically disrupt the interaction with SpoT, but rather acts via destabilisation of the structure of YtfK: attempts to purify the protein failed, (which should be mentioned in the manuscript), and therefore cannot be used as a sufficient specificity control.

AU: This comment confused us. Indeed we already addressed this interesting point in the discussion of our previous submitted version (see copy below).

“The lack of functional interaction between RelA and YtfK is further supported by the observation that YtfK does not affect the level of (p)ppGpp under amino acid starvation, a signal well known to trigger (p)ppGpp-synthesis by RelA¹ (Figure 3C). However we observed that YtfK triggers a stronger accumulation of (p)ppGpp when overexpressed in WT compared to the Δ relA strain (Figure 5A). Moreover ectopic expression of ytfK reduced the growth rate of WT even more tightly to that observed in Δ relA strain (Figure 5D). Therefore RelA also participates to the accumulation of (p)ppGpp promoted by YtfK. Even if we cannot rule out a direct activation of RelA by YtfK under these conditions we suggest that this effect is rather indirect by residual activation of RelA. Indeed, it is well described that the hydrolysis activity of SpoT is required for balancing (p)ppGpp level in the presence of RelA, and disruption of the spoT gene in *E. coli* is therefore lethal². Moreover, it is currently admitted that RSHs avoid to simultaneously synthesize and degrade (p)ppGpp primarily to prevent futile cycle³. Thus, we suggest that YtfK specifically pushes the catalytic balance of SpoT toward (p)ppGpp synthesis rather than hydrolysis therefore enabling residual activation of RelA to maximize the alarmone production. Therefore, the regulation of SpoT activities by YtfK points to an additional layer of regulation to the current stringent response model.”

Finally and following the recommendations of the referee 2 we have purified the 6His recombinant N-terminal part of RelA (RelA¹⁻³⁹⁶) in the same way as SpoT¹⁻³⁷⁸. We performed additional *in vitro* binding assay with SpoT¹⁻³⁷⁸ and YtfK but also with RelA¹⁻³⁹⁶ and YtfK. Under the tested conditions we observed that RelA¹⁻³⁹⁶ does not interact with YtfK (Supplementary Figure 9 of the revised manuscript). Therefore these results further support the specificity of interaction between YtfK and SpoT observed *in vivo*.

Due to the above-mentioned experimental issues it is absolutely essential to run the enzymatic assays suggested during the first round of revision. I am not satisfied with the response of the authors who despite a direct request from me and the second reviewer, do not perform this simple experiment: just saying ‘It is our sincere opinion that such results are not required for our story’ is not a sufficiently strong argument, especially given the alarming DLS results (see below). The authors have already purified SpoT and YtfK, and RelA purification well-documented and is in no way more challenging than that of SpoT and YtfK. The TLC assay is set up, and all that is needed is adding the substrates and resolving the products. It literally takes one day to test the effect of YtfK on SpoT. Testing the effect on RelA would take a week.

AU: It might be realized that such enzymatic *in vitro* reconstitution sounds much harder than it looks and may require purification of the full length SpoT.

I find it very strange that while the authors performed multiple additional *in vitro* experiments (DLS), but decline performing a simple, direct, crucial enzymatic test. As far as I am concerned, this remains an absolutely essential experiment – especially given that now we all agree that the current data can be interpreted that YtfK activates RelA as well.

AU: We do not fully agree with the referee on this point. As mentioned in the discussion we indeed agree that **RelA participates** to the accumulation of (p)ppGpp promoted by YtfK. However even if we cannot totally rule out a direct activation our additional *in vitro* binding assay with RelA (supplementary Figure 9 of the revised manuscript) further support an indirect activation.

Biochemical experiments became especially essential given that the authors themselves (Sinha et al. ‘Fatty acid starvation activates RelA by depleting lysine precursor pyruvate’) have recently shown that a cerulenin challenge causes amino acid starvation leading to RelA-dependent accumulation of ppGpp (note that the authors extensively use cerulenin in their experiments, Figures 3 and 4).

Note also that Sinha et al. (<https://www.biorxiv.org/content/10.1101/635748v1>) is not cited in the paper. It absolutely should be discussed since this result is crucial for interpreting the experiments. Here is a quote from the abstract: ‘Here, we discover that FA starvation leads to rapid activation of RelA and reveal the underlying mechanism. We show that fatty acid starvation leads to depletion of lysine that, in turn, leads to the accumulation of uncharged tRNA^{lys} and activation of RelA. SpoT was also activated by fatty acid starvation but to a lower level and with a delayed kinetics.’

We disagree with the reviewer. As mentioned in our previous response to the referee, the work by Sinha et al., mechanistically showed that indeed fatty acid starvation leads to depletion of lysine that, in turn, leads to the accumulation of uncharged tRNA^{lys} and activation of RelA⁴. However, as mentioned in the work by Sinha et al., this activation **only occurs** in absence of amino acid. Indeed addition of the 20 amino acid or the lysine alone (or pyruvate, the precursor

of lysine) is sufficient to abrogate activation of RelA under FA starvation⁴. Given that our experimental setup include all 20 amino acid we do not believe that such activation occurs in our conditions.

Finally, note that the RelA-dependency under fatty starvation (and in absence of amino acid) is not entirely novelty. Below is a quote from the seminal work by Seyfzadeh et al.,⁵.

“It should be noted that fatty acid starvation might also cause ppGpp accumulation via a relA-dependent mechanism, perhaps indirectly by somehow causing deficiency of some amino acids. Accumulation of ppGpp induced by cerulenin was greater in the wild-type (MG1655) strain than the Δ relA spoT+ strain unless all 20 amino acids were supplied in minimal glucose medium.”

Figure 4: I am unsure that the results on tolerance are warranted in the main text and in the abstract: the loss of YtfK does not have a strong phenotype under clinically-relevant antibiotic challenges (e.g. ampicillin challenge), and there is an effect only with combinatorial knock outs (YtfK and RelA) or clinically-irrelevant combinatorial antibiotic challenges (ampicillin + cerulenin). Stating that ‘YtfK contributes to antibiotic tolerance’ is borderline misleading: while the authors can, indeed, find a clinically irrelevant condition generating a phenotype, this does not advance our understanding of biology. I again strongly recommend moving this figure to the supplement and removing the statement from the abstract: there is no point over-heating the ppGpp vs antibiotic tolerance field.

AU: we followed the recommendation of the referee and moved the figure to the supplementary information and statement has been removed from the abstract. Description in the results section has been condensed to the following 3 lines sentence “Finally and consistent with the proposed role of (p)ppGpp as factor contributing to antibiotic tolerance, we observed that cerulenin pretreatment renders WT and Δ relA mutant cells 10,000-fold more tolerant to ampicillin but fails to substantially protect the Δ ytfK mutant cells.”

Note that recent work from Petra Levin group showed that triclosan (an other well known inhibitor of fatty acid synthesis) promotes antibiotic tolerance. Importantly triclosan-mediated antibiotic tolerance also requires ppGpp synthesis⁶. According to this work approximately 75% of adults in the United States have detectable levels of Triclosan in their urine and 10% of them at concentration equal or above the MIC for *E. coli*⁶. Therefore condition of fatty acid starvation associated to antibiotic treatment might not be totally clinically-irrelevant combinatorial conditions.

The next point is related to the role of ppGpp in antibiotic tolerance. It would be judicious for the authors to remove the citation to their retracted paper on the topic: 34. Germain, E., Roghanian, M., Gerdes, K. & Maisonneuve, E. Stochastic induction of persister cells by HipA through (p)ppGpp-mediated activation of mRNA endonucleases. Proc. Natl. Acad. Sci. U. S. A. 112, (2015).

AU: The reference has been removed accordingly

The newly added DLS experiments show clearly that SpoT does not form a 1:1 complex with YtfK. YtfK itself gives a sharp peak, and if a 1:1 complex would form, the SpoT peak would shift. Instead it becomes more diffuse and heavy-tailed, which strongly speaks against formation of a specific 1:1 complex. What is alarming is in all of the experiments performed in the presence of increasing concentrations of YtfK added to SpoT, there is no separate YtfK peak. This is inconsistent with 1:1 complex formation: at the highest concentration of YtfK (120 μ M), there is a 90 μ M YtfK excess over 30 μ M SpoT. These 90 μ M of YtfK should give

a peak – and there is none. Instead of several well-defined mass peaks corresponding to free YtfK, free SpoT (if 20 uM Kd is correct, there should always be free SpoT) and to YtfK:SpoT peak (poorly resolved with SpoT) there is a broad distribution with a heavy tail, and the tail becomes heavier as YtfK is added. This is all indicative of proteins sticking together non-stoichiometrically, and in the absence of specific controls and enzymatic assays, this invalidates not only the in vitro interactions studies by interferometry – but also suggests that similar non-specific effects are happening in microbiological assays relying on overexpression, thus compromising the rest of the study.

AU: The reviewer says that if there was an interaction between SpoT and YtfK there would be a SpoT peak shift. We have clearly demonstrated in Figure 7, a SpoT peak shift with increasing concentrations of YtfK. This shift is negative rather than positive suggesting that SpoT in presence of YtfK changes its conformational shape to a collapsed globular form rather than an apo-extended form. This negative shift has been shown before in Papish *et al.*⁷ They describe the reduced hydrodynamic radius of calmodulin when in complex with a target peptide. We would like to emphasize that we follow the intensity shifts of SpoT not YtfK. A reminder that particle scattering intensity is proportional to the square of the molecular weight. Given that SpoT is a 43kD protein as a monomer and YtfK an 8kD protein, the presence of free YtfK would be very difficult to detect with this technique, it is for this reason we follow the evolution of SpoT in respect to YtfK. The zoomed-in heavy tail suggested by the reviewer, could be from apo-extended SpoT and not non-specific sticky interaction between YtfK and SpoT. However, we understand the reviewers concern so we have re-analyzed the results using a high resolution method “Multiple Narrow Modes” in the Zetasizer Nano ZS software instead of the custom analysis model for proteins that we used and confirmed absence of aggregation.

To summarise: proper biochemical experiments supported by DLS experiments demonstrating specific 1:1 complex formation between YtfK and SpoT are absolutely essential. Right now the additional experiments only reinforced my concerns.

Minor:

Abstract and Introduction: “synthetase (SYNTH) and hydrolase (HD) motifs” – SYNTH and HD are not motifs. These are either protein domains that perform an enzymatic activity, or in the case of HD a small sequence motif present in metal-dependent phosphohydrolases (please see Aravin and Koonin 1998). The C-terminal half of RSHs proteins is not a domain either – it is a region encompasses several domains. The definition of a domain in structural biology is quite clear: it is part of a given protein that can evolve, function, and fold independently of the rest of the polypeptide. The authors should revise the manuscript throughout in accordance to established concepts and definitions. The statement in the introduction “N-terminal domain of SpoT encompassing the hydrolase and synthetase domains” epitomizes these issues.

AU: We thanks the referee for this important remark. Changes have been made accordingly

Introduction, line 56-57: ‘SpoT has both (p)ppGpp hydrolytic and synthetic activities, similar to RSHs10–12.’ Something is wrong here: since SpoT is an RSH, it cannot be ‘similar to RSHs’. I guess the authors meant Rel, long bi-functional RSH enzyme present in the majority of bacterial species.

AU: The referee is correct. Changes has been made accordingly

BLI experiments: The data was analysed using steady state model, however the sensograms collected for the higher concentration values do not reach steady state (required to report an equilibrium dissociation constant) and it seems that 120 (... 120 seconds? note that the X-axis is not labeled in the figure; it should be) is not enough. Not reaching a clear equilibrium is indicative of experimental setup problems (see above; DLS results). In addition the material methods for this part is lacking, e.g. it is not clear as to how the data was treated to account for the baseline drift and why was the experiment stopped after 90 s.

We have used the Blitz machine from Fortébio to perform the biolayer interferometry experiments. As mentioned in the Material and Methods section the biotinylated Ytfk was bound to a streptavidin biosensor. SpoT¹⁻³⁷⁸ was loaded onto a 4µl drop holder and brought into contact with Ytfk for 90 seconds under agitation at 2200rpm. In all experiments, the BLItz ProTM software performed a reference subtraction of the SpoT¹⁻³⁷⁸ (or RelA¹⁻³⁹⁶) protein response on the uncoated biosensors for each tested concentrations. The maximum contact time on this system is 300 seconds. We have chosen 90 seconds to reduce the protein exposure to ambient temperature. However, even if we wanted to increase the contact time of SpoT¹⁻³⁷⁸ to Ytfk to reach the steady state; it would have needed to equal the dissociation rate and would have taken at least 8 minutes for our highest concentration. We will remove steady state in the results section and add specific binding response at time 110 seconds, which is the sum of 30 seconds baseline + 80 second binding. Most users of Bli technology do not reach steady state, the equilibrium dissociation constants (K_D) are computed from K_{off}/K_{on} as shown is a recent paper in Nature Communications by McLeod, Brandon *et al* ⁸.

Docking: It is not clear how was the docking performed: which program was used? what protein was used as receptor and which as ligand? What score criteria was used for selection? Docking programs do not give a single model, what were the five best model picked up by the docking model and why was this one selected over the others? All these issues should be addressed in the material and methods to be able to reproduce the docking experiment and extract a meaningful insight from it, other this should be removed altogether from the paper.

AU: As mentioned in the previous version of the manuscript (supplementary methods), the docking was performed using HADDOCK2.2 webserver. The input data are:

- YtfK and SpoT models; docking has been performed between two proteins (not protein/ligand).
- The active residues list. Composed by the number of residues that can be involved in the protein-protein interface (the list of chosen actives residue is now provided in Supplementary Table 5 of the revised manuscript).

After docking simulations, Haddock make a clustering and scoring (Haddock score). The scoring is performed according to the weighted sum (HADDOCK score) of the following terms:

- Evdw: van der Waals intermolecular energy
- Eelec: electrostatic intermolecular energy
- Eair: distance restraints energy (only unambiguous and AIR (ambig) restraints)
- Erg: radius of gyration restraint energy
- Esani: direct RDC restraint energy
- Evec: intervector projection angle restraints energy
- Epcs: pseudo contact shift restraint energy
- Edani: diffusion anisotropy energy

- Ecdih: dihedral angle restraints energy
- Esym: symmetry restraints energy (NCS and C2/C3/C5 terms)
- BSA: buried surface area
- dEint: binding energy (Etotal complex - Sum[Etotal components])
- Edesol: desolvation energy

The 10 best clusters are classified according to their Haddock score and a model from each cluster from the best five clusters is now presented in Supplementary Figure 10 and the corresponding cluster parameters/score is now provided in supplementary Table 6 of the revised manuscript.

In the five models YtfK lies at the interface of the Synthetase and Hydrolase domains (supplementary Figure 10)

When presenting the data as a scatter plot (e.g. Figure 3), polynomial fitting in Excel should be avoided – use ‘line and markers’ representation instead. When you fit the data, you subscribe to a certain model that describes the underlying process, be it Hill, exponential decay, Michaelis-Menten. Clearly, the kinetics of ppGpp accumulation is not governed by polynomial equations – and the only function the trace has in this case is a guide for the eye. Connecting the markers with straight lines does exactly that.

AU: The referee is correct. Changes have been made accordingly.

Protein purification: when working with RNA-binding proteins, it is crucial to report the 260/280 ratio for the final preparations. Without these, it is impossible to assess the quality of the preparations. If either SpoT or YtfK comes with RNA, and the other protein has affinity to RNA, this could explain the nonspecific effects observed by DLS.

AU: We agree with the referee. The 260/280 ratio for each protein is now provided in the method section of the revised manuscript.

Line 349: Streptococcus equisimilis – no need to capitalise ‘equisimilis’

AU: We thank the referee and apologize for this typo error. Change has been made accordingly

Reviewer #2 (Remarks to the Author):

General comments

In this revised manuscript, the authors have strengthened their findings that YtfK is an essential factor for the functioning of SpoT in response to phosphate starvation and cerulenin treatment. In particular, they provide new important experiments showing that ytfK is induced in the stress conditions tested. Furthermore, by controlling the level of expression of SpoT and YtfK in vivo, they show that the ratio between the 2 proteins is likely to play a role in SpoT enzymatic regulation.

They have made a good job in responding to most of the reviewers comments. They simply

argue that *in vitro* assay of enzymatic activity is too much to ask at this point given the difficulty to work on these proteins.

AU: We thank the referee for his/her very positive attitude

Concerning the two-hybrid, if it is true that the T25 fusions cannot be detected easily, the sc13582 antibody is very good to detect the T18 fusions. It might have been possible to show the correct production of all the T18 hybrid proteins in general, an especially to compare the T18-YtfK(P42L) and T18-YtfK fusions, because as stated by the authors, there might be a problem of stability of the mutant construct. For the *in vitro* binding study, a control with the N-terminal domain of RelA would still have been an additional strength, maybe with a point mutant in the synthesis domain to help for the expression.

AU: We thank the referee for these very helpful comments. We confirmed that both T18-YtfK and T18-YtfK^{P42L} recombinant proteins were correctly produced as shown by Western blot with the Anti-CyaA (Supplementary Figure 5D of the revised manuscript).

Moreover we have fused SpoT and the truncated variant of SpoT to the T18 fragment to test the interaction with YtfK fused to the T25 fragment and to confirm their production by western blot with the Anti-CyaA (Supplementary Figure 7A-B of the revised manuscript).

Finally we followed the recommendations of the referee and have purified the 6His recombinant N-terminal enzymatic half of RelA (RelA¹⁻³⁹⁶) in the same way as SpoT¹⁻³⁷⁸. We therefore performed additional *in vitro* binding assay with SpoT¹⁻³⁷⁸ and YtfK but also with RelA¹⁻³⁹⁶ and YtfK. Under the tested conditions we observed that RelA¹⁻³⁹⁶ does not interact with YtfK. These results further support the specificity of interaction between YtfK and SpoT observed *in vivo*.

It might be interesting to discuss why the ytfK mutant is not impaired in survival to cerulenin treatment, while it has an effect in stress-induced ampicillin tolerance. This later experiment is in fact the combination of 2 antibiotic treatments, and it might have been more clear to test the ampicillin tolerance with a stress induced by starvation (i.e. phosphate for example).

AU: We agree with the referee. However following the recommendations of the referee 1 this part of the results has been moved to the supplementary file and as suggested has been extensively condensed

Minor comments:

Line 46 : there are more recent articles by the Gourse's lab than the reference given here, describing the 2 binding sites of ppGpp on RNAP.

AU: We apologize. References have been updated

Lines 96 and 99 : I think I understand that the new cloning in pEG25 was to remove the tag, but the reader cannot : explain why a new cloning.

AU: The referee is correct. Change has been done in the main text.

Lines 178-180 : I am confused by the numbers 1,000 fol and 10,000 fold . From what I understand from the figure, the ppGpp^o strain is more affected than the relAytfK mutant.

AU: the referee is correct. We apologize for typo errors. The plating efficiency of the ppGpp⁰ strain drops by more than 100,000-fold (Figure 4A) and the plating efficiency of Δ relA ytfK double mutants drops by more than 10,000 fold after 4h of cerulenin treatment.

Line 268 : typo : oligomerization OF YtfK

AU: We apologize. Change has been made

Line 276 : words missing ? Catalytic « domain presents » conformation changes ?

AU: We apologize. Change has been made

Lines 294-298 : rather than showing that YtfK is involved in antibiotic tolerance, I think that the results shown here are simply « consistent » with the role of YtfK in controlling SpoT and ppGpp levels. And then as a consequence the antibio tolerance because it is already known that ppGpp levels are involved in this phenomenon. I would therefore not say that the results add anything new to the role of SpoT and ppGpp in the control of bacterial multi drug tolerance.

AU: We agree with the referee. Changes have been made according to the suggestion

Line 340 : typo : RSH

AU: We apologize for this typo error. Change has been made

Reviewer #3 (Remarks to the Author):

In the revised manuscript most of the previous concern are addressed or discussed.

AU/ We thank the reviewer for the positive feedback

References

1. Haseltine, W. A. & Block, R. Synthesis of Guanosine Tetra- and Pentaphosphate Requires the Presence of a Codon-Specific, Uncharged Transfer Ribonucleic Acid in the Acceptor Site of Ribosomes. *Proc. Natl. Acad. Sci.* (1973). doi:10.1073/pnas.70.5.1564
2. Xiao, H. *et al.* Residual guanosine 3',5'-bispyrophosphate synthetic activity of relA null mutants can be eliminated by spoT null mutations. *J. Biol. Chem.* (1991).
3. Mechold, U., Murphy, H., Brown, L. & Cashel, M. Intramolecular regulation of the opposing (p)ppGpp catalytic activities of RelSeq, the Rel/Spo enzyme from *Streptococcus equisimilis*. *J. Bacteriol.* (2002). doi:10.1128/JB.184.11.2878-2888.2002
4. Sinha, A. K., Winther, K. S., Roghanian, M. & Gerdes, K. Fatty acid starvation activates RelA by depleting lysine precursor pyruvate. *Mol. Microbiol.* (2019). doi:10.1111/mmi.14366
5. Seyfzadeh, M., Keener, J. & Nomura, M. spoT-dependent accumulation of guanosine tetraphosphate in response to fatty acid starvation in *Escherichia coli*. *Proc. Natl. Acad. Sci. U. S. A.* (1993). doi:10.1073/pnas.90.23.11004
6. Westfall, C. *et al.* The Widely Used Antimicrobial Triclosan Induces High Levels of Antibiotic Tolerance In Vitro and Reduces Antibiotic Efficacy up to 100-Fold In Vivo . *Antimicrob. Agents Chemother.* (2019). doi:10.1128/aac.02312-18
7. Papish, A. L., Tari, L. W. & Vogel, H. J. Dynamic light scattering study of calmodulin-target peptide complexes. *Biophys. J.* (2002). doi:10.1016/S0006-3495(02)73916-7
8. McLeod, B. *et al.* Potent antibody lineage against malaria transmission elicited by human vaccination with Pfs25. *Nat. Commun.* **10**, 4328 (2019).

Reviewers' comments:

Reviewer #2 (Remarks to the Author):

In this revised manuscript, the authors have answered all the reviewers comments, by changing the manuscript organization, by adding an important control experiment with the RelA NTD in the in vitro binding assays, and by adding precise description of the in vitro methods and results. As it stands, the manuscript is concise and well written, the results are very clear and significant. YtfK is a protein required for SpoT-dependent Stringent Response, at least in the two conditions tested, i.e. phosphate and fatty acid starvation. Furthermore, the YtfK/SpoT interaction is proved both by in vivo and in vitro approaches. These are important new findings in the comprehension of ppGpp control in bacteria.

Minor typos

-line 160: synthesis in

-line 270: SpoT1-378

-lines 276-280: "A reminder ... weight" is a sentence without verb, or not finished ? In fact, this small paragraph is not very well written.

-line 284: This shift is negative (remove rather than positive !)

-line 352: in controlling

Reviewer #4 (Remarks to the Author):

The authors aim to determine a direct interaction between SpoT[1-378] and YtfK by using DLS. If SpoT[1-378] and YtfK have a mass of 43 kDa and 8 kDa, respectively, a 1:1 complex will have a mass of 49 kDa.

While it is practically challenging to distinguish between molecules with such a small difference in mass (and hence in size), and it is unlikely that this can be done with a Malvern Zetasizer Nano, it is not theoretically impossible.

However, for small proteins, an analysis in terms of distributions is not appropriate. In this context, the conversion from "intensity" to "volume" distribution is meaningless.

The authors should try to make a proper analysis using single components if they want to give meaning to their experiments.

It is also puzzling the information reported in table S1 concerning a Z-average of 147 nm for YtfK alone. It introduces a warning about the presence of YtfK aggregates at such high concentration, and the consequent recommendation to check if aggregates are also observed in other samples.

In general, these experiments do not demonstrate the authors' claim about a 1:1 interaction. On the other hand, they do not demonstrate the criticism of another referee that the interaction is not specific.

As a matter of fact, they do not demonstrate much, particularly for their inappropriate analysis, and I would suggest to remove them from the manuscript.

We thank the reviewers for critically reviewing the revised manuscript. The comments have helped us to improve the manuscript. We have accommodated all changes. Text modifications are appearing in blue in the core body of the revised manuscript.

With Best Regards

Reviewer #2 (Remarks to the Author):

In this revised manuscript, the authors have answered all the reviewers comments, by changing the manuscript organization, by adding an important control experiment with the RelA NTD in the in vitro binding assays, and by adding precise description of the in vitro methods and results. As it stands, the manuscript is concise and well written, the results are very clear and significant. YtfK is a protein required for SpoT-dependent Stringent Response, at least in the two conditions tested, i.e. phosphate and fatty acid starvation. Furthermore, the YtfK/SpoT interaction is proved both by in vivo and in vitro approaches. These are important new findings in the comprehension of ppGpp control in bacteria.

AU: We thank the referee for his/her very positive attitude. We are very grateful for all valuable comments and suggestions on our manuscript through the several round of revisions.

Minor typos

-line 160: synthesisin in

AU: change has been made

-line 270: SpoT1-378

-lines276-280: "A reminder ... weight" is a sentence without verb, or not finished ? In fact, this small paragraph is not very well written.

-line284: This shift is negative (remove rather than positive !)

AU: Following the recommendation of the referee 4, DLS experiments has been remove from the manuscript, and this paragraph no longer exist in the revised manuscript

-line 352: in controlling

AU: change has been made

Reviewer #4 (Remarks to the Author):

The authors aim to determine a direct interaction between SpoT[1-378] and YtfK by using DLS. If SpoT[1-378] and YtfK have a mass of 43 kDa and 8 kDa, respectively, a 1:1 complex will have a mass of 49 kDa.

While it is practically challenging to distinguish between molecules with such a small difference in mass (and hence in size), and it is unlikely that this can be done with a Malvern Zetasizer Nano, it is not theoretically impossible.

However, for small proteins, an analysis in terms of distributions is not appropriate. In this context, the conversion from "intensity" to "volume" distribution is meaningless.

The authors should try to make a proper analysis using single components if they want to give meaning to their experiments. It is also puzzling the information reported in table S1 concerning a Z-average of 147 nm for YtfK alone. It introduces a warning about the presence of YtfK aggregates at such high concentration, and the consequent recommendation to check if aggregates are also observed in other samples. In general, these experiments do not demonstrate the authors' claim about a 1:1 interaction. On the other hand, they do not demonstrate the criticism of another referee that the interaction is not specific.

As a matter of fact, they do not demonstrate much, particularly for their inappropriate analysis, and I would suggest to remove them from the manuscript.

AU: We are very grateful to the referee and sincerely appreciate his/her critical expertise of our DLS experiments. We paid particular attention to the referee concerns and therefore have followed the recommendation to remove these too preliminary (inconclusive) experiments from the manuscript (including figure 7B and Table S1).